# Defect-gradient-induced Rashba effect in van der Waals PtSe$_2$ layers

Junhyeon Jo [1,6], Jung Hwa Kim[2,6], Choong H. Kim [3,4,6], Jaebyeong Lee[1,6], Daeseong Choe [1], Inseon Oh[1], Seunghyun Lee[1], Zonghoon Lee [1,2✉], Hosub Jin[5✉] & Jung-Woo Yoo [1✉]

Defect engineering is one of the key technologies in materials science, enriching the modern semiconductor industry and providing good test-beds for solid-state physics. While homogenous doping prevails in conventional defect engineering, various artificial defect distributions have been predicted to induce desired physical properties in host materials, especially associated with symmetry breakings. Here, we show layer-by-layer defect-gradients in two-dimensional PtSe$_2$ films developed by selective plasma treatments, which break spatial inversion symmetry and give rise to the Rashba effect. Scanning transmission electron microscopy analyses reveal that Se vacancies extend down to 7 nm from the surface and Se/Pt ratio exhibits linear variation along the layers. The Rashba effect induced by broken inversion symmetry is demonstrated through the observations of nonreciprocal transport behaviors and first-principles density functional theory calculations. Our methodology paves the way for functional defect engineering that entangles spin and momentum of itinerant electrons for emerging electronic applications.

[1] Department of Materials Science and Engineering, Ulsan National Institute of Science and Technology, Ulsan 44919, Republic of Korea. [2] Center for Multidimensional Carbon Materials, Institute for Basic Science (IBS), Ulsan 44919, Republic of Korea. [3] Center for Correlated Electron Systems, Institute for Basic Science (IBS), Seoul 08826, Republic of Korea. [4] Department of Physics and Astronomy, Seoul National University, Seoul 08826, Republic of Korea. [5] Department of Physics, Ulsan National Institute of Science and Technology, Ulsan 44919, Republic of Korea. [6] These authors contributed equally: Junhyeon Jo, Jung Hwa Kim, Choong H. Kim, Jaebyeong Lee. ✉email: zhlee@unist.ac.kr; hsjin@unist.ac.kr; jwyoo@unist.ac.kr

Controlling defects in a pristine material is an alternative method for manipulating physical properties and activating new functionalities of a material. Its outstanding applications, especially by doping, have enriched the modern semiconductor industry and developed a distinct research area, called defect engineering. Generating defects in a target material is commonly implemented through atomic or layered implantation during material growth that guarantees uniform or periodic doping to bulky materials. Beyond atomic implantation, post-treatment, e.g., plasma treatment, is an alternative way for local defect engineering notably in nanoscale materials[1–3]. In particular, its inequivalent reaction from the surface of a binary material could generate selective defect sites. This simple but elaborate manipulation has brought highly improved performance in electronics and photonics with astonishing results such as phase transitions, carrier concentrations, carrier type, etc[4–9].

Spintronics, exploiting both spin and charge of electrons, has been recently benefited from the defect engineering in imprinting a key feature into materials. One of the pivotal goals of defect engineering in spintronic applications is breaking time-reversal symmetry by activating magnetic moments. Intensive efforts over the past decades have been made to develop dilute magnetic semiconductors by integrating random distribution of magnetic dopants[10]. Defect-induced magnetisms in two-dimensional (2D) materials have also been proposed in a number of theoretical studies[11–13]. Various forms of point defects in 2D structures stabilize local magnetic moments, as experimentally shown in such as graphene[14] and transition metal dichalcogenides (TMDs)[15,16]. Another key challenge in defect engineering for spintronics could be inversion symmetry breaking, which leads to the Rashba interaction. The coupling between spin and momentum of itinerant electrons offers an alternative channel for spin sources without ferromagnets. Theoretical studies showed inversion symmetry breaking in 2D materials could be triggered by inducing organized defect distribution[17,18]. However, experimental realization of such periodic defect distribution is unattainable. Thus, it is highly desired to develop a capable approach to break inversion symmetry, which imprints the Rashba interaction in pristine materials and allows electric control of spins in device applications.

2D TMD could be considered as one of the most appropriate platforms for defect engineering due to its ideal crystallinity even in a low dimension. Their various elements and combinations allow to utilize diverse and selective reaction for constitute atoms, especially by plasma treatment[2,9,19]. Mild plasma treatment with Ar gas can make selective vacancies as it etches chalcogen elements, and plasma with $CHF_3$ and $SF_6$ gas pulls out transition metal atoms accompanied by chemical reaction. Further, the energetic reaction of plasma treatment could affect the subsequent TMD layers as dispersing excessive energy. In particular, weak interlayer coupling may allow to withstand layer-by-layer compositional difference. Thus, well-refined plasma treatment with a proper TMD can generate a unique system accommodating a gradient of defects along the layers. As the resulting structural gradient indicates the broken inversion symmetry in the system, an approach to develop a defect-gradient in TMD layers could be the innovative method to activate a Rashba system.

In this study, we demonstrate the generation of a defect gradient and the Rashba effect in 1T-phase $PtSe_2$ thin films through plasma treatment. A structural analysis using scanning transmission electron microscopy shows a defect-gradient along $PtSe_2$ layers in atomic resolution. The Rashba effect induced by the broken inversion symmetry from the defect-gradient is demonstrated by nonreciprocal transport measurement, and first-principles calculations support comprehensive understanding for the defect-gradient and Rashba effect in $PtSe_2$. This study introduces a facile approach to instill the Rashba interaction in layered systems, which couples spin and momentum of electrons for various spin-orbitronic applications.

## Results

**Generation of a defect-gradient in $PtSe_2$.** The host material, van der Waals layered $PtSe_2$ is the best candidate for this study because of its confined and controllable reaction to plasma, a symmetric structure of a pristine film, and compatibility and air stability for device applications. Surface treatments to mechanically exfoliated $PtSe_2$ films were performed through plasma etching with Ar and $SF_6$ mixed gas producing atomically smooth surface modifications (see the Method section and Supplementary Fig. 1). The Ar gas aims for etching of Se atoms, while $SF_6$ chemically reacts with Pt atoms to be vaporized (Fig. 1a). Dispersion of plasma with excessive energy can affect the subsequent TMD layers, which could generate a layer-by-layer gradient defect density, as illustrated in Fig. 1a. Then, a defect-gradient could lead to structural asymmetry, which can derive the Rashba-type spin-splitting in a plasma-treated $PtSe_2$ film. Figure 1b displays the scanning electron microscope image of a studied $PtSe_2$ sample (yellow shaded area) for nonreciprocal magneto-transport measurement. The Rashba spin-orbit interaction significantly affects magneto-transport properties as the effective Rashba field ($B_R$) points the opposite direction for forward and reverse currents ($I_x$), producing nonreciprocal charge transport along the $PtSe_2$

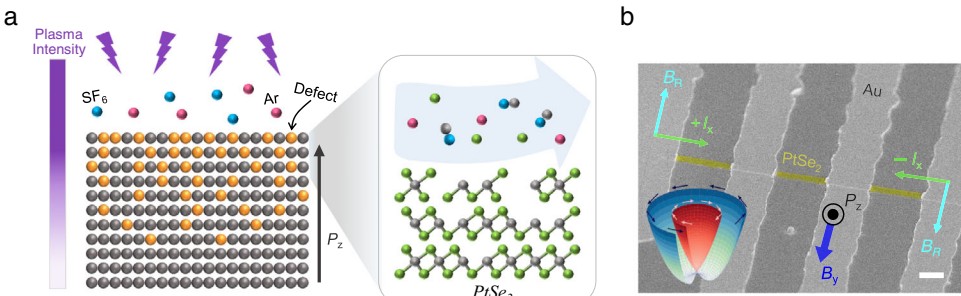

**Fig. 1 Inversion symmetry breaking of $PtSe_2$ layers through a plasma-induced defect-gradient. a** Schematic illustrations of defect-gradient formation on a layered $PtSe_2$ system by plasma treatment. The right sketch shows a process of selective plasma etching of binary elements; Ar for Se and $SF_6$ for Pt. $P_z$ represents the polarization driven by a defect-gradient. **b** A scanning electron microscope image of a fabricated device for the measurement of the nonreciprocal magnetoresistance in a plasma-treated $PtSe_2$ film. The $PtSe_2$ flake (yellow shaded area) is contacted to four Au electrodes (light grey). A scale bar indicates 3 μm. An inset represents a schematic image of Rashba spin-splitting in a plasma-treated $PtSe_2$ film developed by a defect-gradient. $B_R$ represents the effective Rashba field given by the applied currents ($\pm I_x$) and the magnetic field ($B_y$).

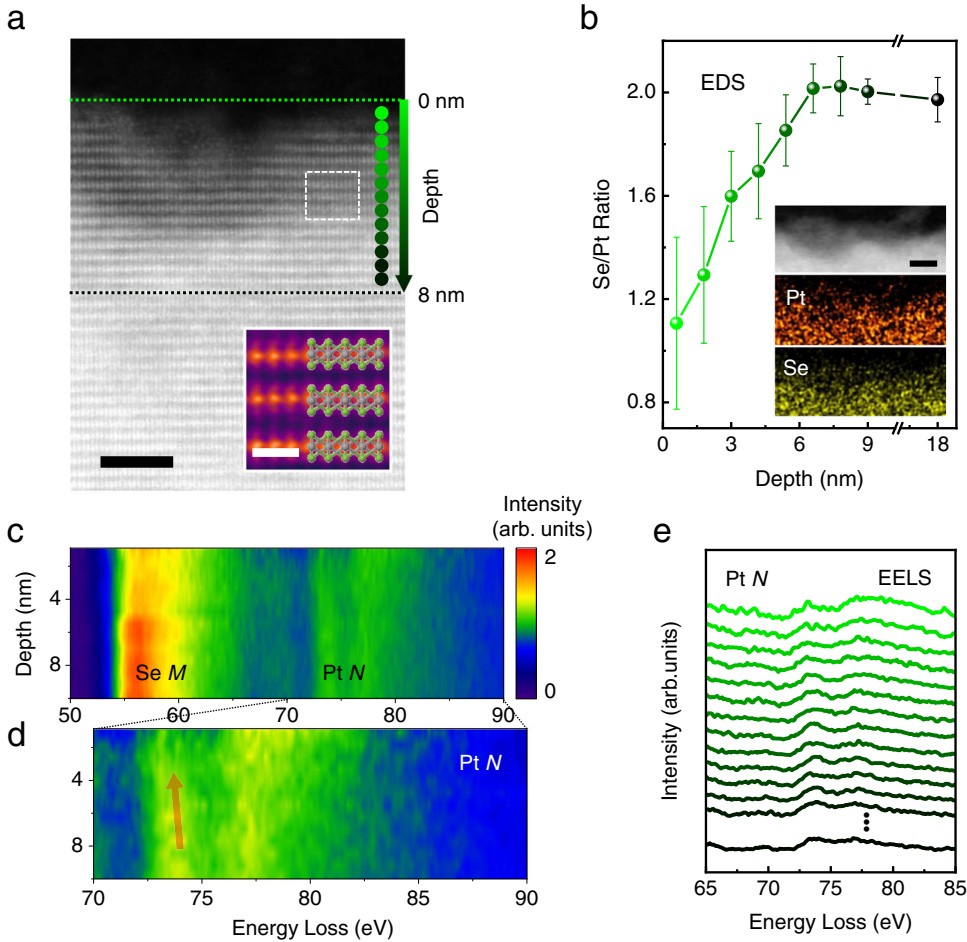

**Fig. 2 Elemental analysis of a plasma-treated PtSe₂ film using atomic-resolution STEM. a** A cross-sectional HAADF-STEM image of plasma-treated PtSe₂ layers. The PtSe₂ layers near a surface exhibit a significant change by plasma treatment. Overlaid circles represent probe positions for STEM spectroscopy analysis. Two dotted lines are guides to the eye indicating the depth from the surface. The inset represents a magnified image of the white dotted box in (**a**) superimposed on a PtSe₂ atomic model. Scale bars for (**a**) and inset are 3 nm and 0.5 nm, respectively. **b** A depth profile of a Se/Pt ratio obtained by EDS measurements. The profile shows a clear gradient of the ratio down to 7 nm from a surface. The error bars indicate the standard deviations of the Se/Pt ratio. Insets are a HAADF-STEM image of a plasma-treated PtSe₂ and its EDS mapping images, showing spatial distribution of Pt and Se near the surface. A scale bars is 5 nm. **c** EELS spectra of Se $M_{4,5}$ and Pt $N_{6,7}$ and **d** magnified Pt $N_{6,7}$ spectra according to the layer depth. EELS data is normalized to the Pt $N_{6,7}$ peak and the scale bar represents a relative intensity. The Pt spectra show a gradual red-shift as approaching to the surface (red arrow). **e** A depth-dependent energy loss profile of Pt $N_{6,7}$. Color contrast of each plot corresponds to the depth location indicated in (**a**).

film. We used mechanically exfoliated PtSe₂ flakes, which had a long rectangular shape along the [100] crystal direction (see Supplementary Fig. 2).

**Structural analysis through atomic resolution STEM.** For the atomic level investigation of a plasma effect on the structural variation in a PtSe₂ film, we employed scanning transmission electron microscopy (STEM). A layer-dependent elemental analysis probed by energy dispersive x-ray spectroscopy (EDS) and electron energy loss spectroscopy (EELS) showed Se vacancies with a steep gradient along the layers. Figure 2a is a cross-sectional high-angle annular dark-field (HAADF) STEM image of a plasma-treated PtSe₂ film. Compared to the deep region of a PtSe₂ film with high crystallinity, plasma-treated PtSe₂ layers near the surface exhibited a decomposed crystal structure. To clarify a change in local environment of Pt, we performed x-ray photoelectron spectroscopy (XPS) analysis for both a pristine film and a plasma-treated PtSe₂ film. A pristine PtSe₂ film exhibited a clear Pt⁴⁺ and Se²⁻ state (Supplementary Fig. 3), but plasma treatment altered some of Pt⁴⁺ elements to Pt²⁺ and Pt⁰ states. A layer-dependent compositional variation was investigated by employing

EDS analysis. Figure 2b shows the observed atomic ratio between Se and Pt according to the depth of the film from the surface. A PtSe₂ layer deeper than 7 nm showed nearly a constant ratio of Se/Pt as 2. However, the Se/Pt ratio steeply decreased down to 1.1 as the layer got close to a surface. These steep variations of the Se/Pt ratio were repeatedly confirmed by analyzing more than 7 different regions (Supplementary Fig. 4). An EDS mapping image also displayed depth-dependent deficiency of Se atoms (the inset of Fig. 2b). EELS data (in Fig. 2c) verifies an atomic gradient of a Se component as compared with Se $M_{4,5}$ and Pt $N_{6,7}$ states at 57 eV and 73 eV, respectively[20]. Since the element ratio is related to the inelastic excitation intensity ratio[21], the decreased Se $M_{4,5}$ intensity compared to the Pt $N_{6,7}$ near a surface region indicates the formation of a defect-gradient from a surface to deep PtSe₂ layers, which is consistent with the EDS result. Additionally, the depth profile of EELS spectra of the Pt $N_{6,7}$ (in Fig. 2d) showed a slightly red-shift of 0.4 eV, and Pt $N_6$ (76 eV) becomes dominant compared to Pt $N_7$ (73 eV) as close to the surface (Fig. 2e). These behaviors can be attributed to the change in the oxidation number of Pt and defect-mediated density of state (DOS) variation because an oxidation state is related to the excitation edge

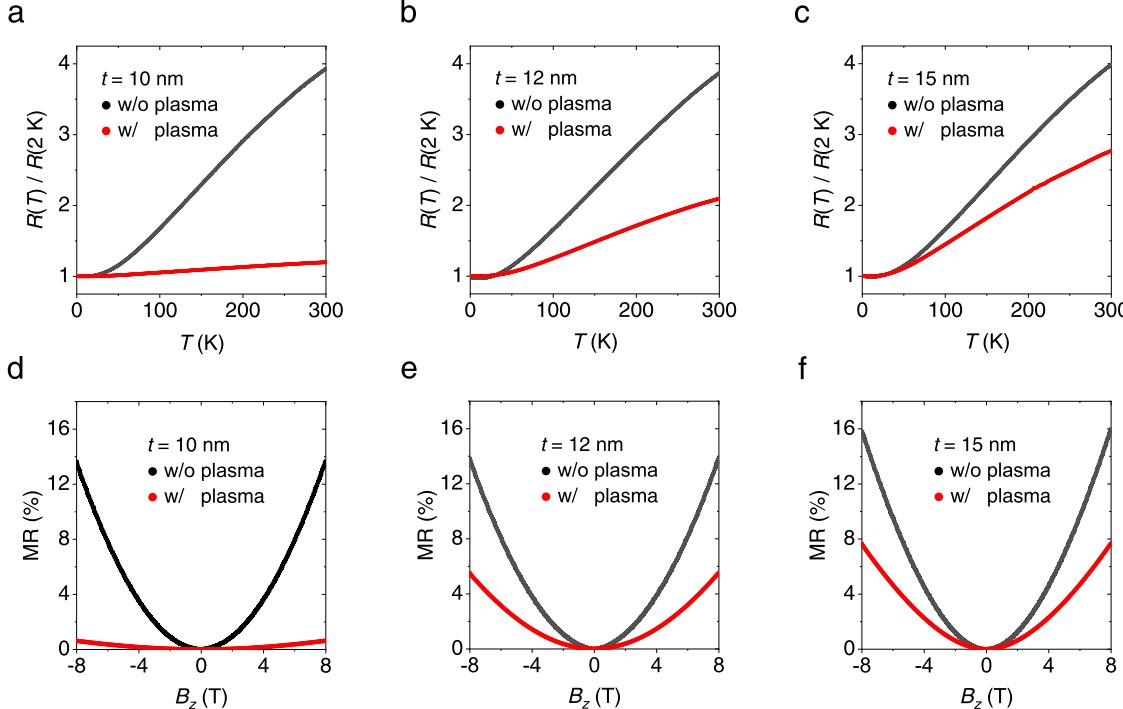

**Fig. 3 Plasma treatment effects on the residual-resistance ratio (RRR) and magnetoresistance (MR) of PtSe₂ upon varying film thickness.** RRR measured for different thicknesses of PtSe₂ films of (**a**) 10 nm, (**b**) 12 nm, and (**c**) 15 nm with and without plasma treatment. Plasma treatment significantly lowers RRR of the PtSe₂, but its effect decreases as the thickness of PtSe₂ increases. MR under a perpendicular magnetic field ($B_z$) measured for different thicknesses of PtSe₂ films of (**d**) 10 nm, (**e**) 12 nm, and (**f**) 15 nm with and without plasma treatment. The MR significantly decreases for the plasma-treated PtSe₂, but its variation reduces with increasing thickness of the film.

shift[22], and the fine structure of core loss EELS is related to the conduction band DOS[23]. In contrast, a pristine PtSe₂ film showed homogenous and well-crystalline structure throughout all regions (Supplementary Fig. 5). Therefore, plasma treatment induces a steep gradient of Se vacancies with a finite skin depth of 7 nm in a PtSe₂ film, establishing a strongly inversion-broken region nearby the surface.

**Magneto-transport analysis for defect-gradient PtSe₂ films**. The formation of defects with a finite depth through plasma treatment can be verified through the residual-resistance ratio (RRR). Figure 3a–c displays temperature-dependent longitudinal resistance ($R_{xx}$) for various thickness of PtSe₂ films with and without plasma treatment. The sample thicknesses of a defect-gradient PtSe₂ (i.e., w/ plasma) indicate a final thickness after plasma treatment. Resistances of the samples increased with increasing temperature regardless of plasma treatment, but the RRR, defined as $R_{xx}$ (300 K)/$R_{xx}$ (2 K), displayed a significant variation. The RRR of pristine PtSe₂ films was higher than 3, whereas plasma-treated PtSe₂ films exhibited much smaller ratio, down to 1.30 (see the values in Supplementary Table 1). Significantly reduced RRR can be attributed to plasma-induced defects in a PtSe₂ film. The variation of RRR due to plasma treatment decreased with increasing the thickness of PtSe₂ (Supplementary Fig. 6a), because the plasma effect had a finite skin depth. The defect-gradient formation also affected to the carrier concentration on PtSe₂ films. The obtained carrier concentrations via Hall measurements were $3.2 \times 10^{21}$ cm⁻³ and $3.6 \times 10^{20}$ cm⁻³ for a pristine PtSe₂ film of 10 nm and a plasma-treated PtSe₂ film of 10 nm at 2 K, respectively (Supplementary Fig. 7). Evidence of induced defects in a PtSe₂ film was also revealed by the magnetoresistance (MR), defined as $(R_{xx}(B) - R_{xx}(B = 0))/R_{xx}(B = 0)$, upon a perpendicular magnetic field ($B_z$) (Fig. 3d–f). The magnitude of ordinary MR

was significantly reduced in a plasma-treated film, because it is proportional to $\sim \mu^2 B^2$ and the carrier mobility $\mu$ is lower for defective samples[24]. Here, the suppression of ordinary MR was also more drastic for a thin PtSe₂ film (Supplementary Fig. 6b), because of the finite depth of the plasma effect.

**Nonreciprocal transport in defect-gradient PtSe₂ films**. The Rashba effect induced by a defect-gradient in a low dimensional system can be probed through transport characteristics. One striking electrical manifestation of the inversion symmetry breaking is the nonreciprocal charge transport as observed in various polar materials[25–30]. Applying a magnetic field in a Rashba system can break the degeneracy between a left and a right mover with opposite spins resulting in directional charge transport. This nonreciprocal response is empirically described as $R = R_0(1 + \gamma(B \times z) \cdot I)$ where $R_0$ is the resistance in a zero magnetic field, $\gamma$ is the coefficient tensor, $z$ is the polarization, and $I$ is the current[25]. Here, $\Delta R = R(I) - R(-I)$ shows maximum magnitude when both the direction of a magnetic field and a current are normal to the electric field due to structural asymmetry. The observation of nonreciprocal charge transport in a single nonmagnetic film reflects the presence of the Rashba spin-orbit interaction, as shown in several noncentrosymmetric systems[25,26,28,30]. Figure 4a displays a schematic illustration for the characterizations of nonreciprocal charge transport. A current was applied in the x-axis and a longitudinal voltage was measured while applying a magnetic field. Figure 4b shows the nonreciprocal MR in a plasma-treated PtSe₂ film (10 nm) in response to the direction of a DC current during a magnetic field sweep in the y-axis. In the presence of large $B_y$, the significant difference of resistance can be observed for a forward and a reverse current flow. For a positive $B_y$, resistance for the forward current flow was higher, while the resistance for the

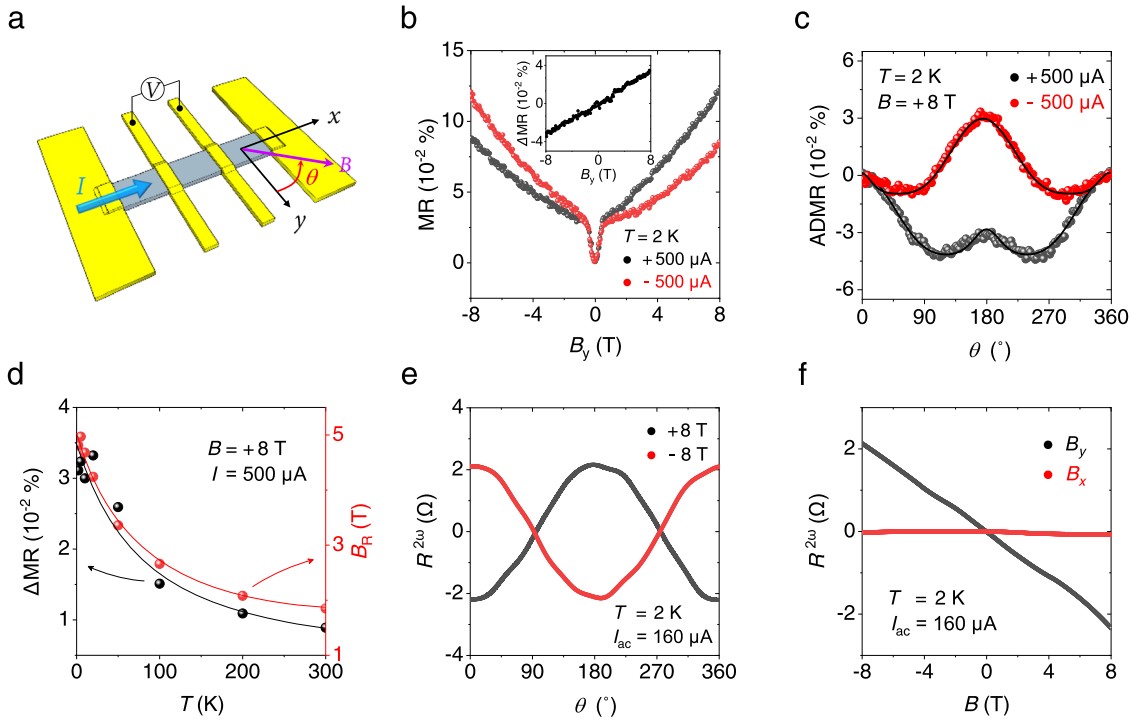

**Fig. 4 Nonreciprocal charge transport in a plasma-treated PtSe₂ film of 10 nm. a** A measurement configuration for the study of the nonreciprocal charge transport. A current ($I$) is applied through the $x$-axis and a voltage ($V$) is measured in a 4-probe configuration under the in-plane magnetic field ($B$). **b** Magnetoresistance (MR) measured with both forward and backward charge flows, $I = \pm 500 \, \mu A$. Results clearly display directional charge transport, following the relation of $\triangle R/R_0 \sim \gamma (B \times z) \cdot I$, where $\triangle R$ is the resistance difference between at a positive and negative current, $R_0$ is the resistance in a zero magnetic field, $\gamma$ is the coefficient tensor, and $z$ is the polarization. An inset displays that the magnitude of nonreciprocal MR linearly increases with increasing the applied magnetic field. **c** Angle-dependent magnetoresistance (ADMR) upon the rotation of a magnetic field in the $yx$ plane with applied currents, $I = \pm 500 \, \mu A$. ADMR exhibits large asymmetry between $\theta = 0\,°\,(+y)$ and $\theta = 180\,°\,(-y)$. This asymmetric behavior is inverted by changing the direction of a charge flow. The solid lines represent the fitted curves for estimating the current-induced Rashba field. **d** Temperature-dependent nonreciprocal $M_R$ and calculated $B_R$ values under $B = +8\,T$ and $I = +500\,\mu A$. Both values decrease as temperature increases, but they persist up to $T = 300\,K$. The solid lines are guides for the eye. **e** Angle-dependent $R^{2\omega}$ measured with a rotation of $B = \pm 8\,T$ in the $yx$ plane and an applied AC current $I_{ac} = 160\,\mu A$. The $R^{2\omega}$ exhibits maximum amplitude when the magnetic field is orthogonal to the directions of a current and polarization in this system. **f** $R^{2\omega}$ in response to different directions of a magnetic field. Nonreciprocal MR is detected only in the presence of the $y$-component of a magnetic field.

reverse current flow became higher for a negative $B_y$. This reversed MR feature clearly represents directional charge transport following, $\triangle R/R_0 \sim \gamma (B \times z) \cdot I$. The MR curves did not display any difference according to the direction of magnetic field sweeps (Supplementary Fig. 8), confirming the observed effects did not originate from sample degradation during the measurement. A small MR feature near $B = 0\,T$ represents an anisotropic magnetoresistance (AMR) from Pt-defect-induced magnetism in a thin PtSe₂ film as observed in previous reports[15,16](see further explanation in Supplementary Fig. 9). An inset in Fig. 4b shows the observed nonreciprocal MR, defined as $\triangle MR = MR(+I) - MR(-I)$, is linear to the magnitude of an applied magnetic field. The magnitude of $\gamma$ is estimated to be $5 \times 10^{-2}\,A^{-1}\,T^{-1}$ that is comparable to those observed in Bi helix[31], chiral organic conductor[27], and $\alpha$-GeTe[32]. On the other hand, a pristine PtSe₂ film without plasma treatment did not exhibit any asymmetry in the same MR measurements (Supplementary Fig. 10). In addition to the control MR measurement, we confirmed that both pristine and plasma-treated PtSe₂ films formed the Ohmic contact to Au electrodes (Supplementary Fig. 11), which allowed us to exclude a contact effect on our nonreciprocal results.

The observed nonreciprocal MR could be simply understood as a result of the current-induced effective Rashba field, which is normal to a current flow in a polar system, as illustrated in Fig. 1b.

Estimation of this effective Rashba field ($B_R$) can be done by measuring angle-dependent MR (ADMR) in the $yx$ plane. Figure 4c shows the measured ADMR, defined as $(R_{xx}(\theta) - R_{xx}(\theta = 0))/R_{xx}(\theta = 0)$, while rotating a device in the $yx$ plane. When a positive current was applied, ADMR at $+y$ ($\theta = 0°$) was higher than ADMR at $-y$ ($\theta = 180°$), but with a negative current, ADMR at $+y$ was lower than ADMR at $-y$. This asymmetric behavior is equivalent to the nonreciprocal MR shown in Fig. 4b. The magnitude and direction of the effective magnetic field ($B_{eff}$) under the rotation of an applied magnetic field ($B_a$) are given as $B_{eff}^2 = B_a^2 + B_R^2 + 2B_a B_R \cos\theta$ and $\alpha = \arctan\left[B_a \cos\theta/(B_R + B_a \cos\theta)\right]$, respectively. Then, ADMR is proportional to $cB_{eff}\cos^2\alpha$, where $c$ is constant[33,34]. Here, an estimated Rashba field was 4.81 T for $I_x = +500\,\mu A$ at $B = +8\,T$. Further measurements with varying currents showed that the current-induced effective Rashba field was linearly proportional to applied currents (Supplementary Fig. 12). The observed nonreciprocal MR even persisted up to room-temperature (Fig. 4d and Supplementary Fig. 13). The directional transport behaviors were also studied for various thicknesses of plasma-treated PtSe₂ thin films (Supplementary Figs. 14–17). The magnitude of nonreciprocal MR tended to decrease with increasing the thickness of the PtSe₂ thin film from 10 nm to 15 nm. This tendency could be attributed to decreasing the effective degree of the Rashba effect in a whole film due to the finite depth of a plasma effect. When the thickness of PtSe₂ decreased below 8 nm, the

material evolved into a semiconducting phase[35,36] (Supplementary Fig. 18). In this regime, the observed nonreciprocal MR decreases with decreasing thickness (Supplementary Figs. 14 and 15).

We further investigated on the nonreciprocal MR using an AC current ($I = I_0 \sin\omega t$) with phase-sensitive detection. In the same geometry displayed in Fig. 4a, a nonreciprocal voltage in response to an AC current for a magnetic field angle $\theta$ can be described as $V^{2\omega}(t) = \gamma R_0 B I_0 \sin\theta \sin\omega t \times I_0 \sin\omega t = \frac{1}{2}\gamma R_0 B I_0^2 \sin\theta\{1 + \sin(2\omega t - \frac{\pi}{2})\}$. Here, a nonreciprocal resistance can be directly detected from the out-of-phase component, as $R^{2\omega} = \frac{V^{2\omega}}{I_0} = \frac{1}{2}\gamma R_0 B I_0 \sin\theta$[28]. Figure 4e displays angle-dependent $R^{2\omega}$ measured with the applied magnetic field of 8 T at $T = 2$ K. Under the rotation of a magnetic field (+8 T), a sinusoidal resistance curve appeared with the maximum amplitude at $+y$ and $-y$. Applying the opposite field of -8 T resulted in an inverted angle-dependent $R^{2\omega}$ curve representing unidirectional characteristics of MR. Figure 4f shows that $R^{2\omega}$ is responsive only to the $y$-component of a magnetic field with linear dependence. $R^{2\omega}$ also exhibited linear dependence on applied currents following $\triangle R/R_0 \sim \gamma(B \times z) \cdot I$ (Supplementary Fig. 19). In short, our magneto-transport study clearly showed all the characteristics of nonreciprocal charge transport, which is the signature of the Rashba effect induced by a defect-gradient in a PtSe$_2$ system.

**First-principles DFT calculation for a defect-gradient PtSe$_2$.** In consistent with the nonreciprocal transport measurements, density functional theory (DFT) calculations clearly show that asymmetric distribution of the Se vacancy in PtSe$_2$ layers gives rise to a large Rashba spin-splitting at the Fermi level. To describe the experimental situation, we prepared a slab geometry that was composed of 10 layers of PtSe$_2$ and 20 Å of vacuum as shown in Fig. 5a. By adopting the virtual crystal approximation in our calculations (see Methods), only the topmost 5 layers contain Se vacancy and the other layers are a pristine region without any vacancy in the PtSe$_2$ film. By interpolating the result of STEM analysis in Fig. 2b, the amount of Se vacancy was the largest at the first layer and linearly decreased to zero below the sixth layer in our simulations. Figure 5b, c, shows a DFT band structure without and with spin-orbit coupling (SOC), respectively. Colors in each band denote the Se/Pt ratio of the Bloch wavefunction; the darker color indicates the larger amount of Se vacancies. In the absence of spin-orbit coupling, each band is doubly degenerated, and no spin-splitting appears as shown in Fig. 5b. Once spin-orbit coupling is turned on, the bands fall into two different categories (Fig. 5c). The dark-colored bands originating from the Se-deficient region show large Rashba-type spin-splitting, whereas the bright ones from a pristine PtSe$_2$ region still maintain the spin degeneracy. Electrons residing in the Se-deficient region only feel the inversion-breaking field caused by a vacancy-gradient. Estimating the Rashba coefficient of the band at the Fermi level yields 2.2 eVÅ along the ΓK line in the Brillouin zone. This value is comparable to that found in giant Rashba materials such as BiTeI (3.8 eVÅ)[37] or (MA)PbI$_3$ (1.4~1.5 eVÅ)[38]. Pristine PtSe$_2$ layers do not show spin-splitting regardless of spin-orbit coupling (Supplementary Fig. 20). Therefore, both spin-orbit coupling and the inversion-breaking field generated by a defect-gradient are essential ingredients that make Rashba-type spin-splitting.

## Discussion

In summary, we successfully demonstrated the generation of a defect-gradient and a resulting Rashba effect in van der Waals 2D layered PtSe$_2$. Optimal plasma treatment with Ar and SF$_6$ gases enabled a selective etching process, which induced a defect-gradient from the surface. STEM analysis revealed that the induced defect-gradient was dominated by Se vacancies and the gradient along layers was formed down to 7 nm depth. Broken spatial inversion symmetry by the defect-gradient brought out the Rashba effect in the PtSe$_2$ film that was evidenced in directional charge transport. In particular, the observed nonreciprocal charge transport persisted even up to room-temperature, which can be utilized for functional two-terminal devices. Theoretical calculations clearly showed the emergence of the Rashba-type spin-splitting by disposing a gradient of Se vacancies along PtSe$_2$ layers. Our study suggests that selective plasma treatment to binary elemental TMD layers could be a facile methodology to develop a defect-gradient along the layers, which could be a new platform to activate the Rashba system for various electronic and spintronic applications. This approach would be also applied for other van der Waals 2D materials standing for a novel and state-of-art technology in the defect engineering.

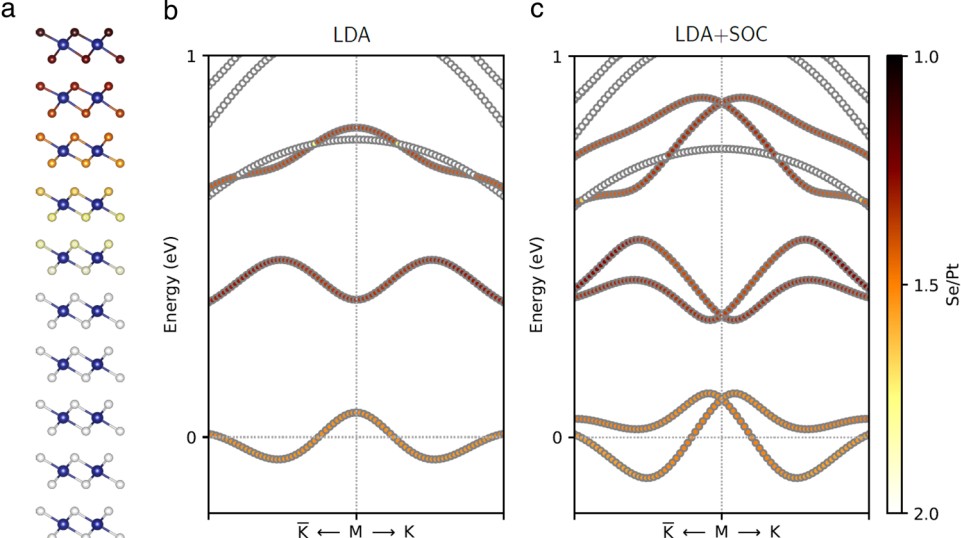

**Fig. 5 First-principles DFT band structure calculations. a** Slab geometry with 10 layers of PtSe$_2$. Blue atoms indicate Pt and the amount of Se vacancy is represented through a color scheme in the right corner of (**c**). Electronic band structures near M point in the Brillouin zone without (**b**) and with (**c**) spin-orbit coupling. Colors in each band indicate the Se/Pt ratio of the Bloch wavefunction.

## Methods

**Sample preparation.** $PtSe_2$ thin films were prepared through mechanical exfoliation from a bulk $PtSe_2$ (purchased from HQ Graphene). $PtSe_2$ films exfoliated on polydimethylsiloxane (PDMS) were transferred onto substrates such as $p$-$Si/SiO_2$ (300 nm), $\alpha$-$Al_2O_3$, and TEM grid, depending on analyses. A reactive ion etching (Labstar, TTL) process with Ar and $SF_6$ mixed gases was performed to the prepared $PtSe_2$ film. Various amounts of the mixed gas, up to 100 sccm depending on conditions, were used with a same ratio of Ar and $SF_6$. The etching decreased the thickness of a $PtSe_2$ film with the etching rate of 0.03 nm/s. Roughness and thickness of each sample were confirmed by using AFM (Veeco) measurement before and after an etching process. The initial thickness of a $PtSe_2$ flake to prepare a plasma-treated $PtSe_2$ is usually 20 nm to 30 nm. This thickness should be, at least, 7 nm-thicker than a target thickness after plasma treatment, to make a 7 nm defect-gradient in a $PtSe_2$ film. Then, identical structures and results can be obtained for the same target thickness of $PtSe_2$ films regardless of their initial thicknesses. XPS (Thermo Fisher) analysis was conducted with a K-alpha spectrometer for an 0.9 mm diameter spot in a UHV chamber ($<10^{-10}$ Torr).

**STEM analysis.** A cross-sectional STEM specimen was prepared by a focused ion beam (Helios 450HP FIB) with a carbon coating to protect the surface of the specimen during ion milling. STEM data was obtained using aberration-corrected FEI Titan Cubed G2 60-300 STEM and all measurements were performed at 200 keV. The convergence angle of the STEM incident beam was 26.6 mrad with a probe current of ~60 pA. An EDS line profile and a spectrum-imaging process were performed using a Super-X system attached to the STEM. EELS spectra were recorded by a Gatan Quantum 965 dual EELS system attached to the STEM. EELS measurements were performed with an energy resolution of 1.0 eV in a 0.1 eV/ channel energy dispersion. Dual EELS measurements were used to simultaneously acquire both the zero-loss and core-loss EELS spectra at each spot to compensate for energy drift during spectrum acquisition.

**Magneto-transport measurement.** $PtSe_2$ flakes were mechanically exfoliated for the transport study. Long and rectangular shape of $PtSe_2$ flakes less than 3 μm width was used for a transport device, because the narrow flakes were highly stable during device fabrication and measurements compared to wide $PtSe_2$ flakes. Four-terminal contacts on a prepared $PtSe_2$ film were patterned by using photo-lithography. Ti (5 nm) and Au (40 nm) were deposited on the electrode patterns in a UHV chamber ($<10^{-8}$ Torr). The deposition chamber had a long distance (~700 mm) between a source and a sample to avoid unwanted damage to samples. All transport measurements were performed in a physical property measurement system (PPMS, Quantum Design) with a horizontal sample rotator. Keithley 2636 A sourcemeter and Keithley 2182 nanovoltmeter were used for DC measurement. For AC measurement, Keithley 6221 was used to supply an AC current with 10 Hz and the AC signal of the first- and the second-harmonic voltage was detected by SR830 DSP lock-in amplifier (Stanford Research).

**First-principles calculation.** Density functional theory calculations were carried out within the local-density-approximation (LDA) with fully relativistic spin-orbit coupling using VASP code[39,40]. A plane wave cut-off energy of 500 eV and $7 \times 7 \times 1$ k-points was used for all calculations. Virtual crystal approximation (VCA) was used to describe the vertically-varying distribution of Se vacancy[41]. In VCA, we constructed the thin film using virtual atoms averaged by Se and vacancy, finally producing the defect distribution shown in Fig. 2b. The VCA has been successfully applied to studies on, for example, topological phase transitions that require very precise handling of structural symmetry and SOC[42–45]. An in-plane lattice constant was fixed at the experimental value of $PtSe_2$. Internal atomic positions were fully relaxed until the maximum force was below 5 meVÅ$^{-1}$ while the interlayer distances were constrained to their experimental value during the relaxation for stability of the calculations. Qualitatively equivalent results were confirmed even if the interlayer distance was fully relaxed. To obtain the average value of the Se/Pt ratio of the $n^{th}$ Bloch state $|\psi_{n\mathbf{k}}\rangle$, we calculated the inner product of the Bloch wavefunction with each atomic orbital, $\mathcal{P}_{\mathbf{k}}^{\alpha} = \sum_{lm} \left| \langle Y_{lm}^{\alpha}, |, \psi_{n\mathbf{k}} \rangle \right|^2$, where $Y_{lm}^{\alpha}$ is the spherical harmonics centered at ion index $\alpha$ and $l, m$ are the angular moment and magnetic moment quantum numbers, respectively. By using the linearly-interpolated value of the Se/Pt ratio in each atom site ($\omega^{\alpha}$), the average value of the Se/Pt ratio of the $n^{th}$ Bloch wavefunction at $\mathbf{k}$ is given by $\mathcal{W}_{n\mathbf{k}} = \sum_{\alpha} \omega^{\alpha} \mathcal{P}_{n\mathbf{k}}^{\alpha}$.

## Data availability

All data are available in the main text or the Supplementary Information. Additional data related to the findings of this study may be requested from the authors.

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

## Acknowledgements

This work was supported by the National Research Foundation of Korea (NRF) grant funded by the Korea government (MSIT) (2020M3F3A2A03082444, 2021R1A2C1008431, and 2020R1F1A1070279). This work was also supported by the Ulsan National Institute of Science and Technology (No. 1.200095.01). J. Jo acknowledges the support of Basic Science Research Program through the NRF funded by the Ministry of Education (2020R1A6A3A03039086). J.H.K. and Z.L. acknowledge the support of Institute for Basic Science (IBS-R019-D1). H.J. acknowledges the support of the National Research Foundation of Korea (NRF) grant funded by the Korea government (MSIT) (2021M3H4A1A0305486411, 2019R1A2C1010498 and 2017M3D1A1040833). C.H.K. was supported by the Institute for Basic Science (IBS-R009-D1)

## Author contributions

J.J. and J.-W.Y. conceived and designed the project. J.J. and J.L. fabricated experimental devices. J.H.K. and Z.L. did TEM/STEM/EDS measurement and analysis. J.J. and J.L. performed magneto-transport measurement and analysis, and S.L., D.C., and I.O. assisted the experiment. C.H.K. and H.J. did density functional theory calculations. J.-W.Y. supervised this project. All the authors participated in the discussion and the preparation of a manuscript.

## Competing interests

The authors declare no competing interests.
