## [Peer Review File · Nature Communications]

Defect-gradient-induced Rashba effect in van der Waals PtSe₂ layersREVIEWER COMMENTS

Reviewer #1 (Remarks to the Author):

The manuscript by J. Jo et al., focuses on the magneto-transport properties of Se-defect dominated PtSe₂ films. The observed directional charge transport up to room temperature is associated with Rashba fields induced as a result of a vertical defect gradient. The manuscript is well-written, and the results are discussed carefully. Literature is well documented. Here, it is also important to state that PtSe₂ has been recently attracting a lot of attention due to its unique thickness-dependent electronic, optoelectronic and induced magnetic properties. Therefore, I believe this manuscript could be interesting to the community focusing on metallic transition metal dichalcogenides if improved. I feel that this work is premature for publication in Nature Communications in its current state. My comments:

- The authors achieve a defect gradient with a dept of ~ 7 nm. In order to fully take the advantage of the gradient, it is important to characterize devices with thicknesses of ≤ 7 nm. In the main text, they only presented devices having PtSe₂ thicker than 10 nm. The observation of a different MR shape for the thickness of 8 nm itself suggests that the work should be extended to $2.5 \text{ nm} < t < 7 \text{ nm}$ regime (< 2.5 nm was reported to be semiconducting).
- It is not clear if the devices labelled as w/ plasma and w/o plasma in Fig.3 are the same devices. If yes, the presented results in Fig.3 would be expected since the thickness reduction due to plasma treatment would affect the thinnest material most as a result of PtSe₂'s unique thickness-dependent transport properties (A. Ciarrocchi et al., Nat. Commun (2018): device resistance drops more rapidly as thickness is reduced). Can they provide more information about their experimental methodology? E.g. what were the initial thicknesses of each crystal shown in Fig3, before plasma treatment?
- If the authors are performing the plasma treatment before fabricating contacts, their contact/PtSe₂ interface will not be the same as its pristine counterpart. Therefore, contact interface related effects could alter its transport properties. For the completeness of their work, I recommend authors perform similar directional magneto-transport measurements in a pristine device, followed by repeating the same measurements on it after creating the defect gradient.
- Observation of a nonreciprocal MR up to RT is indeed interesting. It will be great if they could fully characterize a device at room temperature, as a function of applied current and magnetic field (direction-dependent).
- I think the present title is also a bit misleading. van der Waals nature of PtSe₂ is not used in this work. I recommend the authors change the title to: Se-defect-gradient-induced Rashba effect in PtSe₂ crystals. Just a suggestion.
- What is the origin of the MR signal near $B = 0$ T (Fig 4-b)? It seems that part of the signal disappeared in thicker crystals and became stronger in the thinner, 8 nm sample. Could it be related to the MR signals reported in Pt defect dominated PtSe₂ devices (Ref 15)?

Reviewer #2 (Remarks to the Author):

Controlling defects in crystals is an effective technique for manipulating the physical properties and functionalities. In this manuscript, authors demonstrate that it can be also useful for symmetry engineering, providing a new direction of defect engineering. They report that layer-by-layer defect gradients in two dimensional PtSe₂ give rise to the spatial inversion symmetry and resultant

Rashba effect, which was sensitively probed by nonreciprocal transport measurement, an intrinsic rectification effect reflecting the broken inversion symmetry. They also found that nonreciprocal transport survives up to room temperature, which is scientifically interesting and important for the application.

I feel that results are great and worth publishing in Nature Communications. However, I think the authors should further consider and clarify the following point before accepting the manuscript.

1. I would like to know the effect of carrier number change by defect engineering. In addition to the symmetry change, carrier number change cannot be avoided, which affects the magnitude of the nonreciprocal transport (Nat. Phys. 13, 578 (2017), Nat. Commun. 10, 4510 (2019)).
2. What is the carrier number and expected Fermi level (EF) in each samples?
3. I am interested in whether spin-orbit coupling (SOC) other than Rashba effect can emerge in this system. What is the (effective) point group of this system? Are there any Zeeman-type SOC in such as 2H-type transition metal dichalcogenides? (For example, out-of-plane components of SOC can be probed by applying the magnetic field perpendicular to the two-dimensional materials. (Sci. Adv. 3, e1602390 (2017), Phys. Rev. Lett. 120, 266802 (2018)))
4. There is a kink structure in magnetoresistance (Fig. 4 b, Fig. S13). What is the possible origin of this strange behavior?
5. In 8 nm thick film, magneto resistance (MR) is negative, while the other samples show the positive MR. in 8nm film. What is the difference?
6. In thin film, sub-band structure can emerge, which also affects the nonreciprocal transport (Nat. Commun. 10, 4510 (2019), Sci. Adv. 6, eaay9120 (2020)). I would like to hear authors' opinion on that.

Reviewer #3 (Remarks to the Author):

This manuscript presents an investigation of the potential for engineering a Rashba field within a 2D material using a vertical gradient in the density of structural defects. The material of choice is PtSe₂, and the authors verify the creation of the structural defects with a combination of electron microscopy and spectroscopy. The impact of the potential Rashba field is measured using longitudinal magneto-transport, and the authors demonstrate the onset of an asymmetry in the MR as a function of current direction in samples with defect gradients that is not present in pristine samples. A theoretical basis for this effect is presented using a combination of referencing to prior literature and the presentation of DFT calculations of pristine and defected material.

The material characterization is extremely thorough and the authors demonstrate the creation of the defect gradient that they are attempting to engineer in a very compelling fashion. Similarly, the presence of the asymmetry in magnetoresistance is also clear and indexing to control samples provides a compelling argument that it is related to the defect incorporation. The theoretical foundations for assigning this MR to a Rashba field was less clear to me, and I am unsure if this is simply an issue of clarity in the manuscript or if there are more fundamental issues. As a result, while I believe there is the potential for impactful science here I would like to see a revised manuscript that addresses the following points before advocating for publication.

1. There is some confusion regarding the orientation of the respective fields. First, it is my understanding that the native spin orbit coupling (SOC) in PtSe₂ is out of plane (in the z direction according to the authors convention). Since this is the same direction as the Rashba field proposed as arising from the defects I would appreciate some discussion of how these two effects are distinct and/or can be disentangled. Second, the labeling in the SEM image of their device in Fig 1 is confusing. The image is zoomed in so that only a series of stripes is visible, which I take to be the transport channels of the PtSe₂. Under this convention the authors appear to be proposing a transverse current while their measurements are longitudinal. I think a much more careful description of the sample geometry and the symmetry of the relevant effects is warranted.

2. The role of the DFT in validating the proposed mechanism is not clear to me. The two plots presented are DFT with and without spin orbit coupling, but as noted above PtSe₂ has native SOC that must be taken into account. Are the authors explicitly taking into account different levels of defect incorporation into account in their calculations? If so, there is a high computational cost for the more heavily defected layers and I would be interested to hear more about how they manage this calculation. If not, then the applicability of this DFT work is not clear. How does it support the hypothesis that the defect gradient is giving rise to a spin-orbit field with the appropriate symmetry to generate the experimental results? If the authors are just manually tuning the strength of SOC to mimic what they expect to happen with the different defect densities then this needs to be clearly stated and the implications of this approach need to be more thoroughly discussed.

In summary, this is some very thorough experimental work with a clear signature of current asymmetry, but the discussion of possible mechanisms does not allow for a careful evaluation of whether the author's proposed model is credible. In the absence of such a theoretical understanding I think the impact of the work may be somewhat more narrow than is appropriate for publication in Nature Communications, so this question needs to be resolved prior to publication.

Reviewer #1 (Remarks to the Author):

Reviewer's Comments:

The manuscript by J. Jo et al., focuses on the magneto-transport properties of Se-defect dominated PtSe₂ films. The observed directional charge transport up to room temperature is associated with Rashba fields induced as a result of a vertical defect gradient. The manuscript is well-written, and the results are discussed carefully. Literature is well documented. Here, it is also important to state that PtSe₂ has been recently attracting a lot of attention due to its unique thickness-dependent electronic, optoelectronic and induced magnetic properties. Therefore, I believe this manuscript could be interesting to the community focusing on metallic transition metal dichalcogenides if improved. I feel that this work is premature for publication in Nature Communications in its current state. My comments:

Our Response:

We greatly appreciate the reviewer's important comments as well as positive remarks on our manuscript. We believe that our approach through a defect-gradient would be one of the novel methods to induce inversion symmetry breaking, in particular, in PtSe₂ as well as 2D layered materials that represent unique and superior properties throughout various scientific realms. The following is detailed discussions and corrections we made in response to the reviewer's comments to improve the quality of this manuscript.

#1. Reviewer's Comments:

The authors achieve a defect gradient with a dept of ~7 nm. In order to fully take the advantage of the gradient, it is important to characterize devices with thicknesses of ≤ 7 nm. In the main text, they only presented devices having PtSe₂ thicker than 10 nm. The observation of a different MR shape for the thickness of 8 nm itself suggests that the work should be extended to $2.5 \text{ nm} < t < 7 \text{ nm}$ regime (< 2.5 nm was reported to be semiconducting).

Our Response:

We appreciate the reviewer's comment and suggestion. We agree that the characterization of a thin plasma-treated PtSe₂ device (less than 7 nm) gives us further understanding about the relation between a defect-gradient and its impact on the Rashba effect. Thus, we fabricated a plasma-treated PtSe₂ film of 6 nm through the same method and device structure as described in the main text. Then, we characterized its properties through magnetoelectrical measurements as described below.

We first performed the basic electrical characterization of a plasma-treated PtSe₂ of 6 nm device. A temperature-dependent resistance result in Fig. R1a shows a semiconducting behavior as temperature increases, a measured resistance decreases. It is a different from a metallic behavior in plasma-treated

PtSe₂ films thicker than 10 nm (in the main text). Figure R1b supports the transition showing a Schottky contact at low temperature, in contrast to the Ohmic contact in all plasma-treated PtSe₂ devices thicker than 10 nm in all temperature regions (see our response to comments #3). These results indicate the transition of a plasma-treated PtSe₂ from metal to semiconductor as the thickness of PtSe₂ decreases near 6 nm (Fig. R1c). Note that the metal to semiconductor transition in a pristine PtSe₂ film near 8 nm was also reported [*Adv. Mater.* **29**, 1604230 (2017); *Nat. Commun.* **9**, 919 (2018)].

Fig. R1 Electrical characterization of a plasma-treated 6 nm PtSe₂ film. a RRR of the device. **b** *I*-*V* characterization on various temperatures. **c** Temperature-dependent resistance of plasma-treated PtSe₂ films from 6 nm to 15 nm.

Then, we analyzed a nonreciprocal MR feature in the same device, a plasma-treated PtSe₂ of 6 nm. Figure R2a,b displays the results by sweeping a magnetic field and an angle, respectively. Here, we used the same measurement condition (i.e. the direction *x*, *y*, and *z*) with the geometry in Fig. 4a (in the main text). Both results clearly show the nonreciprocal features in response to the polarity of a current and a magnetic field. Also, the analyzed magnetic field dependence for the nonreciprocal MR shows a linear behavior (the inset in Fig. R2a). These features follow well the nonreciprocal relation $\Delta R/R_0 \sim (B \times z) \cdot I$. Compared to the plasma-treated PtSe₂ films thicker than 8 nm, the nonreciprocal MR (ΔMR) of this 6 nm device showed two orders of magnitude lower value. Following the nonreciprocal relation, the ΔMR values in Fig. R2c

show a linear behavior on applying currents. Further measurement with angle-dependent MR (ADMR) in Fig. R2b exhibits an asymmetric curve and different MR value at $\theta = 0$ (+y direction) and $\theta = 180$ (-y direction). Both the ADMR as well as nonreciprocal MR nearly disappear above 200 K (Fig. R2d).

Fig. R2 Nonreciprocal features of a plasma-treated 6 nm PtSe₂ film. **a** MR measured with currents $I = \pm 500 \mu\text{A}$ at 10 K. The inset indicates $\Delta\text{MR} = \text{MR}(I = +500 \mu\text{A}) - \text{MR}(I = -500 \mu\text{A})$ **b** ADMR measured with the opposite direction of currents under $B = +8$ T at 10 K. **c** Current-dependent ΔMR at 10 K. **d** ADMR measured at various temperatures from 4 K to 300 K.

In response to the reviewer's comments and suggestion, we added the nonreciprocal MR results of a 6 nm plasma-treated PtSe₂ (Supplementary Information Fig. 14) and revised the manuscript in the main text (page 9).

#2. Reviewer's Comments:

It is not clear if the devices labelled as w/ plasma and w/o plasma in Fig.3 are the same devices. If yes, the presented results in Fig.3 would be expected since the thickness reduction due to plasma treatment would affect the thinnest material most as a result of PtSe₂'s unique thickness-dependent transport properties (A. Ciarrocchi et al., Nat. Commun (2018): device resistance drops more rapidly as thickness is reduced). Can they provide more information about their experimental methodology? E.g. what were the initial thicknesses of each crystal shown in Fig.3, before plasma treatment?

Our Response:

We appreciate the reviewer for pointing out the important labeling related to our experiment. The devices labelled as w/ plasma and w/o plasma in Fig. 3 were different devices. The thickness written in the text indicates the final thickness for a plasma-treated PtSe₂ and for a PtSe₂ without plasma treatment, which is mandatory because the resistivity of PtSe₂ significantly varies with thickness as the reviewer mentioned.

First, for “ $t = 10$ nm w/o plasma”, we exfoliated a 10 nm PtSe₂ flake on PDMS and transferred on an α -Al₂O₃ substrate. Then, we made electrode contacts on top of the flake. Here we did not use any plasma treatment and there was no defect-gradient in the PtSe₂ film.

Second, for “ $t = 10$ nm w/ plasma”, we exfoliated PtSe₂ flakes thicker than 20 nm on PDMS and transferred on an α -Al₂O₃ substrate. After that, we etched the flake down to 10 nm. The target thicknesses of films were controlled by plasma time, and the typical etching rate was 0.03 nm/s. Then, electrodes were made on top of the flake. Here, the initial thickness for a plasma-treated PtSe₂ flake should be at least 7 nm-thicker than a target thickness to make a 7 nm defect-gradient in a PtSe₂ film. For example, the initial thicknesses of “ $t = 10$ nm w/ plasma” for SO-11 and SO-12 samples were 27 nm and 19 nm, respectively (see the bottom updated Table R1 for every sample w/ plasma). Even though we used several PtSe₂ flakes having different initial thicknesses (before plasma treatment) to make the same target thickness, the fabricated samples showed almost the same behavior and thickness tendency in RRR and nonreciprocal MR.

With Plasma treatment						
#	B_R (T)	Nonreciprocal MR_y (10^{-2} %)	RR (300K/2K)	MR_z (%)	t (nm)	
					Before plasma etching	After plasma etching
SO-01	1.01	3.9	3.15		25	8
SO-11	4.81	3.2	1.20	0.6	27	10
SO-12	2.98	3.6	1.30	1.3	19	10
SO-21	0.11	0.7	2.12	5.5	45	12
SO-22	0.08	0.6	1.92		35	12
SO-31	0.19	1.0	2.46		30	15
SO-32	0.18	1.1	2.80	7.6	37	15
SO-33	0.19	0.7	2.40		30	15

Table R1 Summary of magneto-transport results measured in plasma-treated PtSe₂ samples and the thickness information before and after plasma treatment.

As the reviewer mentioned, thickness-dependent transport properties of PtSe₂ is unique, and we would like to note that our plasma-treated PtSe₂ films also exhibited this thickness tendency (i.e. the resistivity of PtSe₂ increases with decreasing thickness shown in Fig. R1c). In particular, we confirmed that the transition to a semiconducting behavior in a plasma-treated 6 nm PtSe₂ film (Fig. R1a,c). Our plasma-treated PtSe₂ films also exhibited the same thickness dependence as observed in pristine PtSe₂ films [*Adv. Mater.* **29**, 1604230 (2017); *Nat. Commun.* **9**, 919 (2018)].

In response to the reviewer's comments, we added the detailed thickness information of the studied PtSe₂ films (Supplementary Table 1), resistance results (Supplementary Fig. 18) and further explanation about the device labelled w/ plasma and w/o plasma (page 7 and page 12) in the revised text.

#3. Reviewer's Comments:

If the authors are performing the plasma treatment before fabricating contacts, their contact/PtSe₂ interface will not be the same as its pristine counterpart. Therefore, contact interface related effects could alter its transport properties. For the completeness of their work, I recommend authors perform similar directional magneto-transport measurements in a pristine device, followed by repeating the same measurements on it after creating the defect gradient.

Our Response:

We appreciate the reviewer's comment and suggestion for the contact between a PtSe₂ flake and Au electrodes. Our nonreciprocal MR comes from charge transport through a PtSe₂ longitudinal channel, located in the middle of the source and drain contact, which responds to an applied current and magnetic field. Then, the contact between a PtSe₂ film and Au electrode could contribute to the response to a current and a magnetic field during charge injection/detection through their interface. Measured current-voltage

(I - V) curves in our devices showed the Ohmic contact behavior in both a pristine PtSe₂ device (w/o plasma) and a plasma-treated PtSe₂ device (w/ plasma), shown in Fig. R3, which assured us reciprocal charge injection/detection that did not provide an external nonreciprocal factor to the MR measurement.

Fig. R3 I - V characteristics of the 10 nm PtSe₂ films of a pristine (w/o plasma) and plasma-treated PtSe₂ (w/ plasma).

The measurement the reviewer suggested is an ideal method to exclude any unexpected contact issues. However, if we first measure a pristine 10 nm PtSe₂ device and perform plasma treatment to the pristine PtSe₂ film to make a defect-gradient, the PtSe₂ film will be etched and the thickness will decrease. Thus, we cannot compare a pristine 10 nm PtSe₂ film with a plasma-treated 10 nm PtSe₂ film in a single device. The other method could be that we first exfoliate a 20 nm pristine PtSe₂ film and make electrodes on top of it, which has a contact of pristine PtSe₂/Au electrode. Then, we etch it to make a 10 nm plasma-treated PtSe₂ channel. However, the transport channel between the source and drain electrode will be composed of two different PtSe₂ regions (a pristine PtSe₂ of 20 nm under Au electrodes and a plasma-treated PtSe₂ channel of 10 nm without top Au electrodes), which will definitely affect to overall charge transport.

In response to the reviewer's comments, we added the I - V characteristics of both pristine and plasma-treated 10 nm PtSe₂ films (Supplementary Fig. 11) and further explanation (page 9) in the revised manuscript.

#4. Reviewer's Comments:

Observation of a nonreciprocal MR up to RT is indeed interesting. It will be great if they could fully characterize a device at room temperature, as a function of applied current and magnetic field (direction-dependent).

Our Response:

We greatly appreciate the reviewer's suggestion about room temperature measurement. As the reviewer mentioned, a nonreciprocal MR at room temperature is an attractive feature in particular to extend our approach to practical device application. Thus, we fabricated a plasma-treated PtSe₂ film of 10 nm in the same method and device structure as described in the main text. Then, nonreciprocal MR at 300 K in the fabricated device was measured focusing on angle-dependence and magnetic field-dependence.

First, we measured ADMR at 300 K in the yx plane. Figure R4a-c shows ADMR results depending on a applied current. There was a clear inversion of ΔMR according to the current polarity with $I = +500 \mu A$ and $-500 \mu A$ (Fig. R4a). The magnitude of the ADMR changed as a current increased from $+100 \mu A$ to $+500 \mu A$ (Fig. R4b). These asymmetric MR values (ΔMR), MR difference between the $+y$ direction ($\theta = 0$) and the $-y$ direction ($\theta = 180$), are plotted in Fig. R4c. Switching the polarity of a magnetic field also induced an inversion of a nonreciprocal MR (Fig. R4d). All results obtained for a plasma-treated 10 nm PtSe₂ at 300 K followed well the nonreciprocal MR relation, $\Delta R/R_0 \sim (B \times z) \cdot I$.

We further measured a magnetic field-dependent nonreciprocal MR at 300 K. Figure R4e shows different MR curves depending on a current of $+500 \mu A$ and $-500 \mu A$, and their different MR values represent a nonreciprocal MR feature, as shown in Fig. R4f. This MR plot also shows linear dependence following the nonreciprocal relation.

Therefore, our plasma-treated PtSe₂ system exhibited a nonreciprocal MR at 300 K, following well the relation of $\Delta R/R_0 \sim \gamma(B \times z) \cdot I$. In addition, as we compared the ΔMR values in our new 10 nm plasma-treated PtSe₂ device with the previous 10 nm plasma-treated PtSe₂ device (Fig. 4d and Supplementary Fig. 13), the observed ΔMR and γ values were nearly identical as 0.01 % and 0.015, respectively, which indicated the reproducibility of our methodology to make a defect-gradient and a resulting nonreciprocal MR.

Fig. R4 Nonreciprocal features of a plasma-treated 10 nm PtSe₂ film at 300 K. **a** ADMR under an angle sweep with the different polarity of a current at +8 T. **b** ADMR curves measured with applying various currents from +100 μA to +500 μA . **c** Current-dependent ΔMR estimated at +8 T. **d** ADMR curves measured with the opposite polarity of a magnetic field. **e** MR measured with the opposite polarity of a current. **f** Magnetic field-dependent ΔMR with +500 μA .

In response to the reviewer's comments, we added nonreciprocal MR data at 300 K (Supplementary Fig. 13) and further explanation (page 9) in the revised manuscript.

#5. Reviewer's Comments:

I think the present title is also a bit misleading. van der Waals nature of PtSe₂ is not used in this work. I recommend the authors change the title to: Se-defect-gradient-induced Rashba effect in PtSe₂ crystals. Just a suggestion.

Our Response:

We are grateful for the reviewer's suggestion for the title. Our goal in this study is to build a layer-by-layer defect-gradient in a thin film system. In this respect, two dimensional van der Waals PtSe₂ is the most proper material because of its binary components and the weak interlayer coupling to make composition difference along the layers. At first, if a target material is composed of elements with strong interlayer bonding like metals or oxides, it cannot sufficiently make a gradient through excessive energy dispersion by surface plasma treatment. If a target material consists of only reactive components (easy to etch), such as graphene and MoS₂, it would be totally etched and could not make a gradient in spite of van der Waals films. Thus, the van der Waals nature of interlayer coupling along with constituents of noble atoms (e.g. Au, Pt, and Ag) is essential ingredients to develop a plasma-induced defect-gradient (i.e. an uniform gradient through a wide range along depth). This is why we would like to use the word "van der Waals" PtSe₂ layers in the title.

#6. Reviewer's Comments:

What is the origin of the MR signal near $B = 0$ T (Fig 4-b)? It seems that part of the signal disappeared in thicker crystals and became stronger in the thinner, 8 nm sample. Could it be related to the MR signals reported in Pt defect dominated PtSe₂ devices (Ref 15)?

Our Response:

We appreciate the reviewer's comments about the important issue on low-field MR. As the reviewer mentioned, a small MR signal near $B = 0$ T is related to a Pt defect-induced magnetism as reported [*Nat. Nanotech.* **14**, 674 (2019); *Nat. Commun.* **11**, 4806 (2020)]. We used a PtSe₂ crystal purchased from HQ graphene that was also the source material for the study of the above two previous reports. The synthetic methods for PtSe₂ crystal from HQ graphene is known to induce slight Pt-vacancy defects which lead to long-range magnetic ordering at low temperature. The following Fig. R5 represents anisotropic magnetoresistance (AMR) signals in our PtSe₂ devices. The AMR signal clearly exhibited kink-like features near the coercive field for forward and backward magnetic field sweeps (the inset in Fig. R5a), which is conventional characteristics of a ferromagnet. We observed this AMR signal for both the PtSe₂ films without (Fig. R5a-c) and with plasma treatment (Fig. R5d-f). This implies that our Se defect-gradient

and induced nonreciprocal transport features are not related with Pt defect-induced AMR. In addition, as the thickness of a PtSe₂ film increased, the AMR signal got smaller and eventually disappeared for the film with thickness over 15 nm, which was consistent with the previous reports.

Fig. R5 AMR signals near $B = 0$ T in various PtSe₂ films at 2 K. a AMR in pristine PtSe₂ films from 8 nm to 12 nm. **b** AMR in a plasma-treated PtSe₂ film from 8 nm to 12 nm.

In response to the reviewer's comments, we added the AMR analysis (Supplementary Fig. 9) and further explanation about AMR (page 8) in the revised manuscript.

In summary, the major changes we made in response to the reviewer's comments are as follows.

1. Change of the Fig. 1b image and detailed explanation for a device geometry in the main text.
2. Addition of Supplementary Fig. 7 for carrier concentration and Fermi level change after plasma treatment.
3. Addition of Supplementary Fig. 9 for AMR features in PtSe₂ films.
4. Addition of Supplementary Fig. 11 for the Ohmic contact behavior between a PtSe₂ film and an Au electrode before and after plasma treatment.
5. Addition of Supplementary Fig. 13 for nonreciprocal MR features at room temperature.

6. Addition of Supplementary Fig. 14 for nonreciprocal MR features of a plasma-treated PtSe₂ of 6 nm.
7. Addition of Supplementary Fig. 18 for temperature-dependent resistance depending on thickness and plasma treatment.
8. Addition of thickness information before and after plasma treatment in Supplementary Table 1 and further explanation for device fabrication in the main text.
9. Addition of Supplementary Fig. 20 for DFT band structures of the pristine PtSe₂ film.

Reviewer #2 (Remarks to the Author):

Reviewer's Comments:

Controlling defects in crystals is an effective technique for manipulating the physical properties and functionalities. In this manuscript, authors demonstrate that it can be also useful for symmetry engineering, providing a new direction of defect engineering. They report that layer-by-layer defect gradients in two dimensional PtSe₂ give rise to the spatial inversion symmetry and resultant Rashba effect, which was sensitively probed by nonreciprocal transport measurement, an intrinsic rectification effect reflecting the broken inversion symmetry. They also found that nonreciprocal transport survives up to room temperature, which is scientifically interesting and important for the application.

I feel that results are great and worth publishing in Nature Communications. However, I think the authors should further consider and clarify the following point before accepting the manuscript.

Our Response:

We greatly appreciate the reviewer's assessment on the novelty and importance of our results. As the reviewer mentioned, defect engineering is an effective technique to manipulate the material's properties which have been used for long time in the semiconductor industry, and it has extended its utility to the nanotechnology and, in particular, the two-dimensional materials. We believe that our approach to utilize defects as a layer-by-layer gradient would be a novel method to induce inversion symmetry breaking and use for practical device application. The following is detailed discussions and corrections in response to the reviewer's comments to improve the quality of this manuscript.

#1. Reviewer's Comments:

I would like to know the effect of carrier number change by defect engineering. In addition to the symmetry change, carrier number change cannot be avoided, which affects the magnitude of the nonreciprocal transport (Nat. Phys. 13, 578 (2017), Nat. Commun. 10, 4510 (2019)).

Our Response:

We appreciate the reviewer's comment on the carrier concentration by defect engineering. We performed Hall measurements to check the carrier concentration for a pristine 10 nm PtSe₂ film (w/o plasma) and a plasma-treated 10 nm PtSe₂ film (w/ plasma) in comparison. The carrier concentration for the PtSe₂ w/o plasma was $3.20 \times 10^{21} \text{ cm}^{-3}$ at 2 K and $1.01 \times 10^{22} \text{ cm}^{-3}$ at 300 K. In contrast, plasma treatment reduced the concentration as $3.6 \times 10^{20} \text{ cm}^{-3}$ at 2 K and $1.47 \times 10^{21} \text{ cm}^{-3}$ at 300 K. Figure R6 shows the temperature-dependent carrier concentration and mobility for the PtSe₂ films of 10 nm w/ and w/o plasma treatment.

Fig. R6 Temperature-dependent carrier concentration and mobility measured for 10 nm PtSe₂ devices with and without plasma treatment. a Carrier concentration and **b** mobility values obtained from Hall measurements from 2 K to 300 K.

In response to the reviewer's comments, we added the information about carrier concentration and mobility (Supplementary Fig. 7) and further explanation for the carrier concentration (page 7) in the revised manuscript.

#2. Reviewer's Comments:

What is the carrier number and expected Fermi level (EF) in each samples?

Our Response:

Following up the above response, we calculated the carrier concentration and Fermi level of 10 nm PtSe₂ films w/ and w/o plasma treatment. In a pristine PtSe₂ film of 10 nm, the Fermi level was estimated as 0.791 eV at 2 K with the carrier concentration of $3.20 \times 10^{21} \text{ cm}^{-3}$. In contrast, the estimated Fermi level of a plasma-treated PtSe₂ of 10 nm is 0.185 eV with the carrier concentration of $3.60 \times 10^{20} \text{ cm}^{-3}$ at 2 K, respectively. It was reported that low carrier concentration and Fermi level were favorable to induce higher magnitude of the nonreciprocal magnetoresistance in the Rashba systems [*Nat. Phys.* **13**, 578 (2017); *Nat. Commun.* **10**, 4510 (2019)]. Compared to these previous reports, our plasma-treated PtSe₂ system has rather higher carrier concentration which could be attributed to the low nonreciprocal coefficient (γ) and nonreciprocal MR.

#3. Reviewer's Comments:

3. I am interested in whether spin-orbit coupling (SOC) other than Rashba effect can emerge in this system. What is the (effective) point group of this system? Are there any Zeeman-type SOC in such as 2H-type transition metal dichalcogenides? (For example, out-of-plane components of SOC can be probed by applying the magnetic field perpendicular to the two-dimensional materials. (Sci. Adv. 3, e1602390 (2017), Phys. Rev. Lett. 120, 266802 (2018)))

Our Response:

We are grateful for the reviewer's comments on the structure of PtSe₂. The pristine PtSe₂ was 1T-phase possessing the D_{3d} point group composed of three atomic layers with Pt and Se atoms (Fig. R7). Importantly, this pristine 1T-phase PtSe₂ preserves inversion symmetry, which doesn't show any spin splitting in its band structure (please see Supplementary Fig. 20). Our defect-gradient PtSe₂ thin films would be away from the inversion-symmetric D_{3d} point group, and once we consider the layer-by-layer variation of the defect concentration, it is effectively close to the C_{3v} point group symmetry. The Zeeman-type SOC occurred in 2H-type TMDC monolayer appears through the combination between the absence of inversion symmetry and the presence of horizontal mirror symmetry; the spin splitting induced by inversion asymmetry shows the out-of-plane spin texture that is protected by horizontal mirror symmetry. In our defect-gradient PtSe₂ thin films, the horizontal mirror symmetry as well as the inversion symmetry is severely broken, (mostly) supporting the Rashba-type SOC and helical spin texture. Furthermore, obviously, we could not detect nonreciprocal MR behavior in our magneto-transport measurement under the perpendicular magnetic field B_z (see Fig. 3d-f in the main text) regardless of plasma treatment. Thus, we can infer that the observed nonreciprocal behavior in plasma-treated PtSe₂ film comes from the Rashba effect induced by a defect-gradient.

Fig. R7 Structures of 1T-phase PtSe₂. a Top view and **b** side view of 1T-phase PtSe₂.

In response to the reviewer's comments, we added the band structures of pristine PtSe₂ film (Supplementary Fig. 20) and further explanation (page 11) in the revised manuscript.

4. Reviewer's Comments:

There is a kink structure in magnetoresistance (Fig. 4 b, Fig. S13). What is the possible origin of this strange behavior?

Our Response:

We appreciate the reviewer's comments about the important issue on low-field MR. As the reviewer mentioned, a small MR signal near $B = 0$ T is related to a Pt defect-induced magnetism as reported [*Nat. Nanotech.* **14**, 674 (2019); *Nat. Commun.* **11**, 4806 (2020)]. We used a PtSe₂ crystal purchased from HQ graphene that was the same material with the study of the above two previous reports. The synthetic methods for PtSe₂ crystal from HQ graphene is known to induce slight Pt-vacancy defects, which lead to long-range magnetic ordering at low temperature. The following Fig. R8 represents anisotropic magnetoresistance (AMR) signals in our PtSe₂ devices. The AMR signal clearly exhibited kink-like features near the coercive field for forward and backward magnetic field sweeps (the inset in Fig. R8a), which is conventional characteristics of a ferromagnet. We observed this AMR signal for both the PtSe₂ films without (Fig. R8a-c) and with plasma treatment (Fig. R8d-f). This implies that our Se defect-gradient and induced nonreciprocal transport features are not related with Pt defect-induced AMR. In addition, as the thickness of a PtSe₂ film increased, the AMR signal got smaller and eventually disappeared for the thicker film over 15 nm, which was consistent with the previous reports.

Fig. R8 AMR signals near $B = 0$ T in PtSe₂ films at 2 K. **a** AMR in pristine PtSe₂ films from 8 nm to 12 nm. **b** AMR in a plasma-treated PtSe₂ film from 8 nm to 12 nm.

In response to the reviewer's comments, we added the AMR results (Supplementary Fig. 9) and further explanation (page 8) in the revised manuscript.

5. Reviewer's Comments:

In 8 nm thick film, magneto resistance (MR) is negative, while the other samples show the positive MR. in 8 nm film. What is the difference?

Our Response:

We appreciate the reviewer's comment about a negative MR under B_y in a 8 nm PtSe₂ film. There are several factors which can induce a negative MR in thin film systems such as weak localization, Kondo effect, chiral anomaly, magnetism, and current jetting.

- Chiral anomaly is one of the representative sources which results in a negative MR in Weyl metals when B is parallel to E [*Phys. Rev. B* **88**, 104412 (2013)]. Recently, this negative MR was reported in a thin PtSe₂ film [*Commun. Phys.* **3**, 93 (2020); *Adv. Funct. Mater.* **31**, 2104192 (2021)]. Our 8 nm PtSe₂ film also displayed a negative MR under B_x (i.e. $B//E$, Fig. R10a) and followed the chiral conductivity (σ) relation which represented the proportional relation between the σ and B^2 (Fig. R9b). In addition, the MR under B_y also displayed a negative MR (Supplementary Fig. 15c and Fig. R9c). However, as this negative MR under B_y and its origin have not been studied yet, it is difficult to conclude the relation between the negative MR under B_y and the chiral anomaly.

- Weak localization can result in a negative MR when a magnetic field is applied to the perpendicular direction (B_z). However, our sample showed the ordinary parabolic behavior of MR under B_z at 2 K and persisted up to $B_z = 8$ T that was far away from the low magnetic field region for weak localization usually less than 4 T. This absence of weak localization effect in our PtSe₂ film can exclude its effect on the negative MR under B_y .

- Magnetic ordering can induce a negative MR, generally we called anisotropic magnetoresistance (AMR) in a ferromagnet. Both 8 nm and 10 nm PtSe₂ films clearly showed small AMR signals, *i. e.* the hysteresis feature during the forward and backward magnetic field sweep (Fig. R8a,b). It was reported that this AMR came from Pt defect-induced magnetism in PtSe₂ [*Nat. Nanotech.* **14**, 674 (2019); *Nat. Commun.* **11**, 4806 (2020)]. Thus, the negative MR shown exclusively for a 8 nm PtSe₂ film can be distinguished from the AMR (Fig. R8d and Fig. R9c), and we can exclude the effect from magnetic ordering.

- Current jetting can be induced by non-homogenous current distribution. It usually occurs in a large sample which does not have a well defined channel and electrodes [*Nat. Commun.* **7**, 11615 (2016)]. Thus, it is not related to our device with a few micron-meter scale electrode patterned by photolithography.

- Kondo effect can be a source for a negative MR in a thin film system. As the Kondo effect comes from the isotropic s - d exchange interaction between conduction electrons and localized magnetic moments, the MR resulting from the Kondo effect should be independent of the magnetic field direction [*Adv. Mater.* **33**, 2005465 (2020)]. In our PtSe₂ system, there is clear difference in MR curves under B_x (Fig. R9b) and B_y (Fig. R9c), even if we consider a nonreciprocal MR feature.

Even though we have checked possible factors for the origin of a negative MR under B_y in a 8 nm PtSe₂ film, we could not clearly conclude the origin. This negative MR under B_y exhibited not only in a plasma-

treated 8 nm PtSe₂ film but also in a pristine 8 nm PtSe₂ film (Fig. R10), which might indicate that a defect-gradient would not play a critical role for a negative MR under B_y . We note that all PtSe₂ films (w/ and w/o plasma) thicker than 10 nm showed a positive MR under B_y (and B_x as well).

Fig. R9 Transport characterization in a plasma-treated 8 nm PtSe₂ film. **a** Device geometry of a measured PtSe₂ film. **b** Magnetic field (B_x)-dependent conductivity. The inset shows the linear relation between the conductivity and B_x^2 . **c** Magnetic field (B_y)-dependent conductivity.

Fig. R10 Conductivity depending on B_x in a pristine 8 nm PtSe₂ film.

#6. Reviewer's Comments:

In thin film, sub-band structure can emerge, which also affects the nonreciprocal transport (Nat. Commun. 10, 4510 (2019), Sci. Adv. 6, eaay9120 (2020)). I would like to hear authors' opinion on that.

Our Response:

As mentioned by the referee, the sub-band structure of the defect-gradient PtSe₂ thin films could play a crucial role in the non-reciprocal transport. According to our DFT calculation in Fig 5c, we have two different types of sub-bands depending on their spatial distribution. The one sub-bands (colored by dark brown) mostly located in the defect-gradient region feel the strong inversion-breaking field, resulting in

the large Rashba-type spin splitting. The other sub-bands (colored by white) located in the pristine region hardly see the inversion-asymmetry, showing no spin splitting.

In summary, the major changes we made in response to the reviewer's comments are as follows.

1. Change of the Fig. 1b image and detailed explanation for a device geometry in the main text.
2. Addition of Supplementary Fig. 7 for carrier concentration and Fermi level change after plasma treatment.
3. Addition of Supplementary Fig. 9 for AMR features in PtSe₂ films.
4. Addition of Supplementary Fig. 11 for the Ohmic contact behavior between a PtSe₂ film and an Au electrode before and after plasma treatment.
5. Addition of Supplementary Fig. 13 for nonreciprocal MR features at room temperature.
6. Addition of Supplementary Fig. 14 for nonreciprocal MR features of a plasma-treated PtSe₂ of 6 nm.
7. Addition of Supplementary Fig. 18 for temperature-dependent resistance depending on thickness and plasma treatment.
8. Addition of thickness information before and after plasma treatment in Supplementary Table 1 and further explanation for device fabrication in the main text.
9. Addition of Supplementary Fig. 20 for DFT band structures of the pristine PtSe₂ film.

Reviewer #3 (Remarks to the Author):

Reviewer's Comments:

This manuscript presents an investigation of the potential for engineering a Rashba field within a 2D material using a vertical gradient in the density of structural defects. The material of choice is PtSe₂, and the authors verify the creation of the structural defects with a combination of electron microscopy and spectroscopy. The impact of the potential Rashba field is measured using longitudinal magneto-transport, and the authors demonstrate the onset of an asymmetry in the MR as a function of current direction in samples with defect gradients that is not present in pristine samples. A theoretical basis for this effect is presented using a combination of referencing to prior literature and the presentation of DFT calculations of pristine and defected material.

The material characterization is extremely thorough and the authors demonstrate the creation of the defect gradient that they are attempting to engineer in a very compelling fashion. Similarly, the presence of the asymmetry in magnetoresistance is also clear and indexing to control samples provides a compelling argument that it is related to the defect incorporation. The theoretical foundations for assigning this MR to a Rashba field was less clear to me, and I am unsure if this is simply an issue of clarity in the manuscript or if there are more fundamental issues. As a result, while I believe there is the potential for impactful science here I would like to see a revised manuscript that addresses the following points before advocating for publication.

Our Response:

We greatly appreciate the reviewer's meticulous assessment and comments for recognizing the novelty and importance of our results. We believe that a defect-gradient is a novel approach to develop a layer-by-layer structure to induce the inversion symmetry breaking and Rashba field. Therefore, we introduced TEM analysis, magnetotransport measurement, and DFT calculation to prove our defect-gradient structure and propose its utility for practical device application and scientific research. The following is detailed discussions and corrections in response to the reviewer's comments to improve the quality of this manuscript.

#1. Reviewer's Comments:

There is some confusion regarding the orientation of the respective fields. First, it is my understanding that the native spin orbit coupling (SOC) in PtSe₂ is out of plane (in the z direction according to the authors convention). Since this is the same direction as the Rashba field proposed as arising from the defects I would appreciate some discussion of how these two effects are distinct and/or can be disentangled. Second, the labeling in the SEM image of their device in Fig 1 is confusing. The image is zoomed in so

that only a series of stripes is visible, which I take to be the transport channels of the PtSe₂. Under this convention the authors appear to be proposing a transverse current while their measurements are longitudinal. I think a much more careful description of the sample geometry and the symmetry of the relevant effects is warranted.

Our Response:

We appreciate the reviewer's comments on the structure of PtSe₂. We used 1T-phase PtSe₂ in this study, possessing the D_{3d} point group composed of three atomic layers with Pt and Se atoms. This implies that the pristine 1T-phase PtSe₂ thin films preserve inversion symmetry and show the spin degenerate band structure. In other words, there is no spin splitting in its band structure. (see Supplementary Fig. 20.) Our defect-gradient PtSe₂ system would be different from the symmetric D_{3d} point group of the pristine, and it is effectively close to the C_{3v} point group, *supporting the effective inversion-breaking electric field perpendicular to the plane*. As a result of this out-of-plane inversion-breaking field, the defect-gradient PtSe₂ thin films can possess the Rashba-type SOC and helical spin texture. The disentanglement of Zeeman type and Rashba type spin-splitting can be probed by applying the magnetic field perpendicular to the two-dimensional materials [*Sci. Adv.* **3**, e1602390 (2017); *Phys. Rev. Lett.* **120**, 266802 (2018)]. However, we could not detect any nonreciprocal MR behavior in our magneto-transport measurement under the out-of-plane magnetic field (Fig. 3d-f in the main text). Thus, we can infer that the observed nonreciprocal behavior originated from the Rashba effect due to a defect-gradient in a PtSe₂ film.

We agree the reviewer's comment on the SEM image. To make it clear, we revised colors, marks, and explanation in Fig. 1b. Here, we would like to mention that we used a large size of Au electrodes because of our plasma treatment process. For the plasma treatment, we used an α -Al₂O₃ substrate that was stable for the etching with a SF₆ gas, in contrast to a conventional Si/SiO₂ substrate that was harshly etched (20 times higher etching rate than PtSe₂). Unfortunately, this α -Al₂O₃ substrate could not be used for our electron beam lithography system. Thus, we used photolithography to make electrode patterns, which made relatively the large width of 4-terminal electrodes and the rough edge of patterns.

In response to the reviewer's comments, we modified Fig. 1b and added Supplementary Fig. 20 with further explanation (page 5, page 8, page 11, and page 19) in the revised manuscript.

#2. Reviewer's Comments:

The role of the DFT in validating the proposed mechanism is not clear to me. The two plots presented are DFT with and without spin orbit coupling, but as noted above PtSe₂ has native SOC that must be taken into account. Are the authors explicitly taking into account different levels of defect incorporation into account in their calculations? If so, there is a high computational cost for the more heavily defected layers and I would be interested to hear more about how they manage this calculation. If not, then the

applicability of this DFT work is not clear. How does it support the hypothesis that the defect gradient is giving rise to a spin-orbit field with the appropriate symmetry to generate the experimental results? If the authors are just manually tuning the strength of SOC to mimic what they expect to happen with the different defect densities then this needs to be clearly stated and the implications of this approach need to be more thoroughly discussed.

Our Response:

Applying the first-principles DFT method to the system with defects requires some approximations for the treatment of the disordered defects. A direct approach is to use a supercell approximation to consider the isolated defect. As the reviewer pointed out, such calculations are computationally very demanding. To circumvent this difficulty, we used virtual crystal approximation (VCA) that is a computationally less expensive approach and at the same time can capture the averaged effect of the disordered defects. In VCA, we construct our thin film geometry with the primitive periodicity using virtual atoms averaged by Se and vacancy, indeed producing the defect distribution shown in Fig. 2b. This technique has widely used in various band-structure calculations. Previous work has demonstrated good accuracy in some semiconductor and ferromagnetic materials (see R1-R7 below). In Ref. R7, in particular, supercell calculation and VCA showed consistent results. In addition to this, VCA has been successfully applied to studies on topological phase transitions that require very precise handling of structural symmetry and SOC [R8-R14]. It seems that VCA is sufficient to examine how the defect qualitatively affects the symmetry of the system and the strength of the SOC, especially in our work.

The Rashba splitting is a combined effect of SOC and inversion-breaking field. The purpose of our calculations is to reveal that Rashba-type spin-splitting occurs due to defect-gradient in our experiments. Therefore, we tried to show that both atomic SOC and defect-gradient are necessary for the finite spin-splitting. First, as shown in Fig. 5b,c, we can confirm that SOC is necessary to generate Rashba-type spin-splitting since we do not see any splitting without SOC in the defect-gradient system. On the other hand, the inversion-breaking induced by the defect-gradient has been also turned out to be an essential ingredient for the spin-splitting (Supplementary Fig. 20). The splitting does not appear regardless of SOC in the inversion symmetric system of a pristine PtSe₂ film.

[R1] S. de Gironcoli, P. Giannozzi, and S. Baroni, Phys. Rev. Lett. **66**, 2116 (1991).

[R2] N. Marzari, S. de Gironcoli, and S. Baroni, Phys. Rev. Lett. **72**, 4001 (1994).

[R3] A.M. Saitta, S. de Gironcoli, and S. Baroni, Phys. Rev. Lett. **80**, 4939 (1998).

[R4] D.A. Papaconstantopoulos and W.E. Pickett, Phys. Rev. B **57**, 12 751 (1998).

[R5] W.E. Pickett and D.J. Singh, Phys. Rev. B **53**, 1146 (1996).

[R6] P. Slavenburg, Phys. Rev. B **55**, 16 110 (1997).

[R7] C. Eckhardt, K. Hummer, and G. Kresse, Phys. Rev. B **89**, 165201 (2014).

[R8] B. M. Wojek, et. al., Phys. Rev. B **87**, 115106 (2013).

[R9] S. Safaei, P. Kacman, and R. Buczko, Phys. Rev. B **88**, 045305 (2013).

- [R10] Weizhao Chen, Yufei Zhao, Qiushi Yao, Jing Zhang, and Qihang Liu, Phys. Rev. B **103**, L201102 (2021).
[R11] N. Mitsuishi, Nat. Commun. **11**, 2466 (2020)
[R12] B. Chen, Nat. Commun., **10**, 4469 (2019)
[R13] G. Xu, B. Lian, P. Tang, X.-L. Qi, and S.-C. Zhang, Phys. Rev. Lett. **117**, 047001 (2016)
[R14] G. W. Winkler, et. al., Phys. Rev. Lett. **117**, 076403 (2016)

In response to the reviewer's comments, we added the band structures for pristine PtSe₂ films (Supplementary Fig. 20), explanation (page 11 and page 14), and further references [42-45] in the revised manuscript.

Reviewer's Comments:

In summary, this is some very thorough experimental work with a clear signature of current asymmetry, but the discussion of possible mechanisms does not allow for a careful evaluation of whether the author's proposed model is credible. In the absence of such a theoretical understanding I think the impact of the work may be somewhat more narrow than is appropriate for publication in Nature Communications, so this question needs to be resolved prior to publication.

Our Response:

We greatly appreciate the reviewer for pointing out our theoretical calculations in this study. We responded the reviewer's each comment, in particular about theoretical approach to be clear. Now, we believe that our comprehensive research through experimental and theoretical methods well explain and demonstrate the Rashba-type SOC system in a defect-gradient PtSe₂ film which would be one particular method to engineer 2D layered materials.

In summary, the major changes we made in response to the reviewer's comments are as follows.

1. Change of the Fig. 1b image and detailed explanation for a device geometry in the main text.
2. Addition of Supplementary Fig. 7 for carrier concentration and Fermi level change after plasma treatment.
3. Addition of Supplementary Fig. 9 for AMR features in PtSe₂ films.
4. Addition of Supplementary Fig. 11 for the Ohmic contact behavior between a PtSe₂ film and an Au electrode before and after plasma treatment.
5. Addition of Supplementary Fig. 13 for nonreciprocal MR features at room temperature.
6. Addition of Supplementary Fig. 14 for nonreciprocal MR features of a plasma-treated PtSe₂ of 6 nm.
7. Addition of Supplementary Fig. 18 for temperature-dependent resistance depending on thickness and plasma treatment.
8. Addition of thickness information before and after plasma treatment in Supplementary Table 1 and further explanation for device fabrication in the main text.
9. Addition of Supplementary Fig. 20 for DFT band structures of the pristine PtSe₂ film.

REVIEWERS' COMMENTS

Reviewer #1 (Remarks to the Author):

In the revised manuscript, the authors did a great work and clarified all (in my opinion) points raised by referees. It is a very thorough study and I highly recommend publishing in Nature Communications.

Reviewer #2 (Remarks to the Author):

Authors answered all the questions in a satisfactory way. I have no additional questions.

Reviewer #3 (Remarks to the Author):

I have carefully reviewed both the author responses to earlier comments and the revised manuscript and believe that this manuscript is now suitable for publication. The authors have done an impressively thorough job of addressing the questions raised by all reviewers and I believe the impact and novelty of their work is now much more evident. I have no further comments.

Please check the items below carefully and add a response in each row of the table to indicate the changes that you have made. Please also check through any additional marked-up edits we may have provided within the manuscript file.

Abstract and editor's summary

Our guidance:

Your response:

Your paper will be accompanied by the following editor's summary. Please let us know if there are any inaccuracies: 'Materials with strong Rashba-type spin orbit coupling hold promise for spintronic applications and the investigation of topological phases of matter. Here, the authors report a method to generate layer-by-layer defect gradients in a van der Waals semiconductor, inducing broken spatial inversion symmetry and Rashba effect in the material.'	We would like to change "semiconductor" to "layered material" because the studied PtSe2 is metallic.
Please replace the second sentence of the abstract with the following: "While homogenous doping prevails in conventional defect engineering, various artificial defect distributions have been predicted to induce desired physical properties in host materials, especially associated with symmetry breakings."	We changed it.

Author information

Our guidance:

Your response:

Please review your complete author list to verify that it is complete and accurate. We ask that you consult with your coauthors to ensure that all names, affiliations, and titles are represented correctly. Note that if any authors are added or removed after this point then all authors will be requested to provide approval documentation that could potentially delay the production of your paper.	We confirmed.
---	----------------------

Article structure

Our guidance:

Your response:

We can accommodate up to 10 display items (Figures or Tables) in the main article. Each Figure and Table must fit easily within an A4 page (210 x 297 mm). Please ensure that the number and	We confirmed.
---	----------------------

size of your Figures and Tables fulfil these requirements to avoid any delay in the acceptance of your article.	
To comply with our article templates, the text must be split into:  - Introduction (<1000 words), which must include the background and rationale for the work. The final paragraph should be a brief summary of the major results and conclusions. The results of the current study should only be discussed in this final paragraph. The Introduction should contain no references to figures or tables. - Results, which must be split into subheaded sections, ensuring that the subheadings are no longer than 60 characters including spaces. Subheadings should contain no punctuation. - Discussion, without subheadings. - Methods, which must be split into subheaded sections, ensuring that the subheadings are no longer than 60 characters including spaces. There is no word limit for this section. 	We changed them.
Please remove all the references to Figures in your Introduction.	We confirmed.
Please ensure your main manuscript file includes the following sections, in this order: Title Author list Affiliations Abstract Introduction Results Discussion (optional) Results and Discussion (optional) Methods (including Data Availability, Code Availability and Statistics subsections where relevant) References Acknowledgements Author Contributions Statement Competing Interests Statement Tables Figure Legends/Captions (for main text figures) We do not edit Supplementary Information files; they will be uploaded with the published article as they are submitted with the final version of your manuscript. Any tracked changes should be removed from the file and the file should be provided as a PDF file. Supplementary Figures do not need to be provided separately.	We confirmed.

Main text

Our guidance:

Your response:

The clarity of your manuscript would strongly benefit from English language editing. We recommend that you either ask a colleague to review your manuscript or that you use one of the many English language editing services available. Two such services are provided by our affiliates Nature Research Editing Service (https://authorservices.springernature.com/go/nr/?utm_source=nroasLetters&utm_medium=email&utm_campaign=natcommsletters) and American Journal Experts (https://www.aje.com/go/natureresearch/?utm_source=nroasLetters&utm_medium=email&utm_campaign=natcommsletters) Nature Communications authors are entitled to a 10% discount on their first submission to either of these services. To claim 10% off English editing from	We confirmed.
---	---------------

Nature Research Editing Service, follow this link: https://authorservices.springernature.com/go/nr/?utm_source=nroasLetters&utm_medium=email&utm_campaign=natcommsletters To claim 10% off American Journal Experts, follow this link: https://www.aje.com/go/natureresearch/?utm_source=nroasLetters&utm_medium=email&utm_campaign=natcommsletters	
Please do not use italics, bold font, underlining or speech marks unless required for technical terms (in both the main text and the display items).	We confirmed.
Please make sure that mathematical terms throughout your manuscript and Supplementary Information (including in figures, figure axes, and legends) conform strictly to the following guidelines. Equations must be supplied in editable format, and not as images. Scalar variables (e.g. x , V , χ) must be typeset in italic, whereas multi-letter variables and functions (e.g. log) must be formatted in roman. Vectors (such as the wavevector k or the magnetic field vector B) must be typeset in bold without italics.	We confirmed.
Please use bold font for numbering chemical compounds, but not for chemical abbreviations or formulae in both the main text and the display items.	We confirmed.
Atomic orbital notations (sp, d, etc.) and corresponding XPS labels should be typeset in italics throughout the main text, figures and Supplementary Information, whereas all accompanying superscripts/subscripts should be typeset in roman font.	We confirmed.

Figures and Tables

Our guidance:

Your response:

Please see the guidelines linked below for detailed instructions about how your figures should be prepared. Following these instructions will reduce the chances of delays should we need to request replacement artwork from you at a later stage. https://www.nature.com/documents/NRJs-guide-to-preparing-final-artwork.pdf	We checked it.
Please make sure that the terms 'atomic units (a. u.)' or 'arbitrary units (arb. units)' are appropriately used.	We changed it.
Any abbreviations, symbols or colours present in your figures must be defined in the associated legends.	We confirmed.
In each Figure and Supplementary Figure where error bars are used, they must be defined.	We confirmed.
In the caption of Fig. 2a please mention the represented horizontal dotted lines.	We mentioned it.
In the caption of Fig. 2b please describe how the represented error bars were obtained.	We described it.
Please do not refer to parts of figures ('top', 'middle', 'left', etc.) in the text. Instead, if necessary, provide panel labels for the parts you wish to refer to. Panels should be individually labelled as "a,b,c,d,...". In particular please	We changed it.

relabel the top and bottom panels of Fig. 2c. Please adapt the related caption and description in the Main Text accordingly.	
Please label the quantity represented by the colour scale in Fig. 3c. In the caption, please also mention the represented orange arrow.	We mentioned them (in Fig. 2c).
In the captions of Fig. 3,4 please define the acronyms RRR, MR and ADMR.	We defined it.
In the caption of Fig. 4, please define V , I , B , θ , ΔR , R_0 , γ .	We changed it.
In the caption of Figs. 4c,d please describe the represented solid lines.	We described it.

Data and Code

Our guidance:

Your response:

Nature journals strongly support public availability of data and code. Please deposit the data and code used in your paper into a public data repository, or alternatively, present the data as Supplementary Information. If data can only be shared on request, please explain why in your Data Availability Statement, and also in the correspondence with your editor. Please note that for some data types, deposition in a public repository is mandatory. Any restrictions on sharing of these data types must be clearly indicated in the statement and discussed with the editor. More information on our data deposition policies and available repositories can be found here: https://www.nature.com/nature-research/editorial-policies/reporting-standards#availability-of-data	We understood.
All published manuscripts reporting original research in Nature Portfolio journals must include a data availability statement, as a separate section before the References and under the heading 'Data Availability'. The data availability statement must make the conditions of access to the “minimum dataset” that are necessary to interpret, verify and extend the research in the article, transparent to readers. This minimum dataset may be provided through deposition in public community/discipline-specific repositories, custom proprietary repositories or general repositories like Figshare, Zenodo and Dryad. Providing large datasets in supplementary information is strongly discouraged and the preferred approach is to make data available in repositories. Scientific Data, a Nature Portfolio journal, maintains a list of approved and recommended data repositories to support researchers seeking suitable repositories for their data (https://www.nature.com/sdata/policies/repositories). The Data Availability Statement should also reference any source data published alongside the paper. If DOIs are provided, we also strongly encourage including these in the Reference list (authors, title, publisher (repository name), identifier, year). For clinical datasets or third party data, please ensure that the statement adheres to our policy (https://www.nature.com/nature-research/editorial-policies/reporting-standards#availability-of-data)	We understood.
Please use the following template to provide all the information stated above: The XX data generated in this study have been deposited in the YY database under accession code ZZ [add hyperlink here]. The XX data are available under restricted access for {insert reason}, access can be obtained by {explain how}. The raw XX data are protected and are not available due to data privacy laws. The processed XX data are available at YY. The XX data generated in this study are provided in the Supplementary Information/Source Data file. The XX	We confirmed

data used in this study are available in the YY database under accession code ZZ [Add hyperlink here].	
--	--

Methods

Our guidance:

Your response:

Sufficient details of the experiments must be provided in the Methods section such that they could be reproduced without reference to published papers. Use of the term "as described previously" is not encouraged.	We confirmed.
--	---------------

End matter

Our guidance:

Your response:

Nature Portfolio defines Competing Interest (CI) as financial and non-financial interests (including but not limited to funding, employment, stocks, shares, patents, personal or professional relationships with individuals or institutions, and unpaid membership advocacy) that could be perceived to directly undermine the objectivity, integrity, and value of a publication, or could be seen as having an influence on the judgments and actions of authors with regard to objective data presentation, analysis, and interpretation. Please thoroughly review our policy on Competing Interests and include a detailed statement both in your final manuscript file and in our manuscript tracking system. Please ensure the statements are identical in both. Be specific about how each point stated relates to the research and list applicable author initials, and/or patent numbers. If there are no competing interests, a negative statement must be included. https://www.nature.com/nature-research/editorial-policies/competing-interests	The authors declare no competing interests.
Please confirm that all relevant funding awarded to each author is described in the Acknowledgements section. List each grant number, followed by the initials of the author who received it.	We confirmed.

Preparing your manuscript files

Our guidance:

Your response:

Unless otherwise stated please limit individual file sizes to approximately 30MB. We strongly encourage the use of repositories for large datasets or source data due to size considerations.	We confirmed.
The use or adaptation of previously published images is strongly discouraged. If this is unavoidable, please request the necessary rights documentation to re-use such material from the relevant copyright holders and return this to us when you submit your revised manuscript. Please check whether your manuscript or Supplementary Information contain third-party images, such as figures from the literature, stock photos, clip art or commercial satellite and map data. For more information on what constitutes ownership by a third party, please contact our Editorial Assistant at naturecommunications@nature.com	We confirmed.

Forms to complete

Our guidance:

Your response:

Editorial Policy Checklist Please update and upload a final version of the Editorial Policy Checklist with your revised manuscript files. A blank Editorial Policy Checklist can be found via the link below. Note that this form is a dynamic 'smart pdf' and must be downloaded and completed in Adobe Reader. Please update your current checklist or download from: https://www.nature.com/documents/nr-editorial-policy-checklist.zip	We did it and uploaded it.

You will need to upload:

Editorial Policy Checklist	We confirmed.
Completed Third Party Rights Table (if relevant)	-
A completed copy of this checklist	We confirmed.
The main article file in Microsoft Word format - please supply a version with tracked changes and a version with tracked changes accepted	We confirmed.
Separate Figure files	We confirmed.
Inventory of Supporting Information	We confirmed.
A Supplementary Information file	We confirmed.